# Review of experimental studies of secondary ice production

Alexei Korolev[1] and Thomas Leisner[2]
[1] Environment and Climate Change Canada, Canada
[2] Karlsruhe Institute of Technology, Karlsruhe, Germany and University of Heidelberg, Germany

*Correspondence to*: Alexei Korolev (alexei.korolev@canada.ca)

### Abstract

Secondary ice production (SIP) plays a key role in the formation of ice particles in tropospheric clouds. Future improvement of the accuracy of weather prediction and climate models relies on a proper description of SIP in numerical simulations. For now, laboratory studies remain a primary tool for developing physically based parameterizations for cloud modeling. Over the past seven decades, six different SIP-identifying mechanisms have emerged: (1) shattering during droplet freezing; (2) the rime splintering (Hallett-Mossop) process; (3) fragmentation due to ice-ice collision; (4) ice particle fragmentation due to thermal shock; (5) fragmentation of sublimating ice; (6) activation of ice nucleating particles in transient supersaturation around freezing drops. This work presents a critical review of the laboratory studies related to secondary ice production. While some of the six mechanisms have received little research attention, for others contradictory results have been obtained by different research groups. Unfortunately, despite vast investigative efforts, the lack of consistency and the gaps in the accumulated knowledge hinder the development of quantitative descriptions of any of the six SIP mechanisms. The present work aims to identify gaps in our knowledge of SIP as well as to stimulate further laboratory studies focused on obtaining a quantitative description of efficiencies for each SIP mechanism.

### 1. Introduction

Secondary ice production (SIP) is defined here as the formation of atmospheric ice as a result of processes involving pre-existing ice particles, in contrast to primary ice production, which commences by the nucleation of ice either homogeneously in strongly supercooled droplets or heterogeneously on the surface of ice nucleating particles (INP) (e.g. Kanji et al, 2017). SIP is one of the fundamental cloud microphysical processes, recognized as a major contributor to the observed concentration of ice particles at temperatures warmer than the homogeneous freezing temperature.

Even though SIP was observed in early laboratory experiments (e.g. Dudetski and Sidorov, 1911, Findeisen, 1940; Findeisen and Findeisen, 1943; Brewer and Palmer, 1949; Malkina and Zak, 1952; Puzanov and Accuratov, 1952; Schaefer, 1952; Bigg, 1957), the geophysical significance of SIP was recognized only after the beginning of regular airborne studies of cloud microstructure in different geographical regions (e.g. Koenig 1963, 1965; Hobbs, 1969; Mossop, 1970, 1985a; Mossop et al. 1964, 1972; Ono, 1972; Hallett et al. 1978; Hobbs and Rangno 1985, 1989; Beard 1992; and many others). A systematically observed enhancement

of the number concentration of cloud ice particles over the concentration of INP in the same air mass urged for the provision of an explanation of the physical processes underlying this discrepancy.

From the late 1950s to early 1970s, six possible mechanisms were proposed explaining the secondary production of ice crystals. However, since then, limited progress has been made in understanding of how each of those mechanisms contribute to the ice particle concentrations and what the necessary and sufficient conditions are for initiating each of these mechanisms. This situation is complicated by the fact that numerical cloud models tend to focus on only one of the six possible mechanisms, namely the rime-splintering (Hallett-Mossop) process, whereas other mechanisms have been disregarded.

Beyond recent reviews on in-situ studies of ice multiplication (e.g. Cantrell and Heymsfield, 2005; Field et al., 2017), little attention has been devoted to exploring the details of laboratory studies on SIP mechanisms. To bridge this gap, this paper provides an extended review of experimental works on SIP. Laboratory studies with reproducible and controlled environments are the basic means of examining physical processes underlying each SIP mechanism, as well as quantifying the rates of secondary ice production, and identifying necessary and sufficient conditions required for initiation of these mechanisms. Without this knowledge, a development of the physically based parameterisations of SIP in weather prediction and climate simulations is not feasible. Due to their coarse spatial and temporal resolution, in-situ airborne (by nature Eulerian) observations should be used for validation and feedback of laboratory and theoretical SIP studies, rather than serve as a primary tool for developing parameterizations for numerical simulations.

This work is an overview of the current knowledge on SIP obtained from laboratory studies. In-situ observations and theoretical studies of SIP are mentioned here occasionally, though many of them remained outside the frame of this review. For the sake of thoroughness, experimental studies of the effects of artificial ice particle fragmentation during in-situ sampling were included in this review as well.

This review aims to provide navigation for future experimental works that seek to enhance our understanding of SIP mechanisms.

The present paper describes laboratory studies of the following SIP mechanisms: the fragmentation of droplets during their freezing (section 2), rime splintering (section 3), fragmentation due to collision of ice particles with each other (section 4), ice particle fragmentation due to thermal shock caused by freezing droplets on their surface (section 5), fragmentation of sublimating ice particles (section 6), activation of ice nucleating particles in transient supersaturation around freezing drops (section 7). Section 8 describes experimental studies that look at spurious enhancement of ice concentration during in-situ measurements, which can be confused with SIP. The concluding remarks are presented in section 9.

The authors would like to acknowledge the length disproportions between the aforementioned sections. Section 2 has the biggest volume, which is a reflection of the large amount on knowledge accumulated on different aspects of water freezing directly linked to the secondary ice formation during droplet freezing. The

rest of the sections are smaller in size due to fewer laboratory experiments related to them. These disproportions will be discussed in more detail in section 9.1.

## 2. Fragmentation of freezing drops

Historically, the first mechanism proposed to explain SIP was the fragmentation of freezing droplets (e.g. Langham and Mason, 1958; Mason and Maybank, 1960; Kachurin and Bekryaev, 1960; Muchnik and Rudko, 1961). During the freezing process of a cloud droplet, liquid water may be trapped inside a growing ice shell formed around the droplet. The expansion of ice during further freezing results in an increase of pressure inside the ice shell. If the pressure exceeds a critical point, the ice shell may crack or shatter to relieve the

internal pressure. The ice fragments that result from droplet cracking or shattering will serve as secondary ice. In addition, gases dissolved in the droplet might be released during the pressure drop events. Gas bubbles may burst upon freezing at the colder droplet surface, resulting in a second source of fresh small ice fragments.

One of the necessary conditions for SIP during droplet freezing is the creation of a closed ice shell and subsequent inward freezing. Therefore, depending on the way in which the droplet freezes, it may or may not

generate secondary ice. Hence, our consideration begins with a review of studies on the process of droplet freezing.

### 2.1 Freezing stages of a supercooled drop

The process of freezing of a supercooled droplet can be divided into two main stages. The first stage is a

90 process that involves negligible heat exchange with the surrounding air. During this period, a dendritic ice network (slushy ice) forms through the liquid phase, releasing latent heat, and heating up the liquid toward the melting point. This stage is usually referred to as the "fast" or "recalescence" stage. The second stage is quasi-isothermal and determined by the freezing of the remaining liquid water. The heat transfer during this stage is directed to the air-droplet interface. The second stage is usually called the "slow" or "freezing" stage. After

95 freezing is complete, the temperature of the frozen droplet gradually decreases towards the ambient temperature to attain a thermal equilibrium.

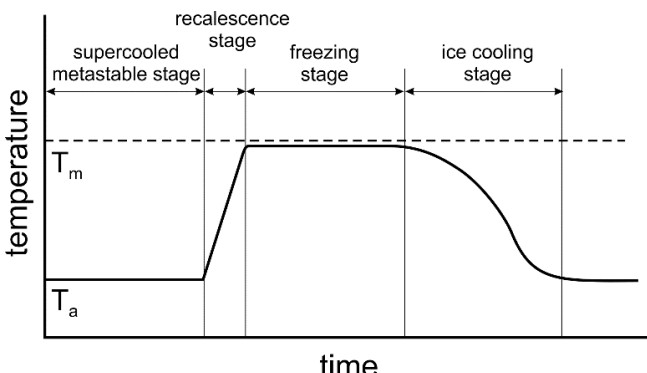

**Figure 1** A conceptual diagram of temperature changes during the freezing of a supercooled droplet. Here $T_m$, $T_a$ are the melting and environmental temperatures, respectively. During the metastable stage droplet temperature is assumed to be
equal to the air temperature $T_a$.

A conceptual diagram of the temperature changes during the freezing of a supercooled droplet is shown in Fig.1a. Documented temperature changes during the freezing of supercooled liquid drops can be found in e.g. Mason and Maybank (1960), Muchnik and Rudko (1961), Pena et al. (1969), Bauerecker et al. (2008), Tavakoli et al. (2015).

## 2.2 Freezing fraction

The amount of frozen liquid water $\Delta m$ during the recalescence stage can be estimated from a simplified equation of heat balance:

$$\Delta m L_m = \Delta m c_i \Delta T + (m - \Delta m) c_w \Delta T + \Delta Q \tag{1}$$

where $m$ is the droplet mass, $\Delta T = T_m - T_a$ is the droplet supercooling; $T_m$, $T_a$ are the melting point and initial droplet temperatures, respectively; $L_m$ is the latent heat of freezing, $c_i$, and $c_w$ are the specific heat of ice and liquid water, $\Delta Q$ is the heat loss due to thermal exchange with the environment. Description of variables is provided in Appendix A. Eq.1 assumes that the droplet nucleating temperature $T_n = T_a$.

After neglecting $\Delta Q$ and $(c_i - c_w)\Delta m \Delta T$, Eq.1 yields an approximation of the fraction of water $\mu = \Delta m/m$ frozen during the recalescence stage as

$$\mu = \frac{c_w \Delta T}{L_m} \tag{2}$$

Down to a temperature of -30°C, Eq. 2 is in a very good agreement with an exact solution of (1) with T-dependent material properties. Using a nuclear magnetic resonance technique, Hindmarsh et al. (2005), measured a fraction of frozen water formed in a supercooled 2mm diameter drop during the recalescence stage of freezing. They found the experimentally measured $\mu$ is in good agreement with that predicted by Eq.2 (Fig. 2).

Equation 2 yields that only a relatively small fraction of water freezes during the first stage. Thus, at -4°C and -20°C, the frozen fraction of water will be approximately 5% and 23%, respectively.

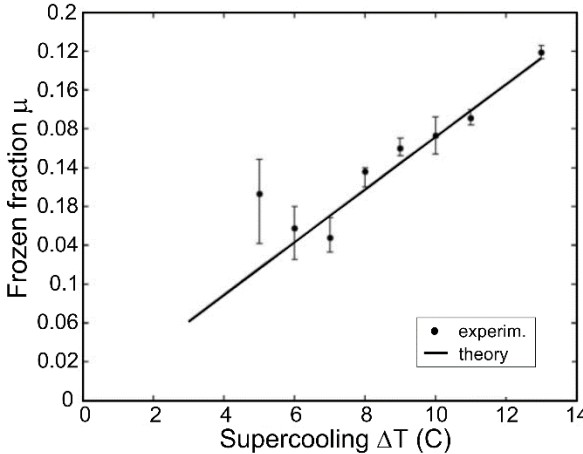

**Figure 2.** A frozen fraction of water $\mu$ formed in a 2mm diameter drops during the recalescence stage versus temperature. The experimentally measured $\mu$ is in good agreement with that theoretically predicted by Eq.2 (adapted from Hindmarsh et al. 2005)

### 2.3 Droplet freezing time

The time scale of the recalescence stage can be assessed as (Macklin and Payne, 1967)

$$t_1 = D/G(\Delta T) \tag{3}$$

where $D$ is the droplet diameter, and $G(\Delta T)$ is the rate of ice growth at water supercooling $\Delta T$. The growth rate $G(\Delta T)$ was studied by many research groups (e.g. Lindenmeyer et al. 1959; Hallett, 1964; Pruppacher, 1967a; Feuillebois, et al. 1995; Shibkov et al. 2003, 2005 and others). It was found that the velocity of ice growth

along the $c$-axis $G_c$ is considerably smaller than that along the $a$-axis $G_a$ (e.g. Macklin and Ryan, 1968). The velocity of freely growing ice is determined as $G = (G_a^2 + G_c^2)^{1/2}$. The summary of studies of the velocity of freely growing ice as a function of $\Delta T$ is shown in Fig.3.

Following Fig.3 and Eq.3 at $T_a$ =-4°C and -20°C the recalescence time $t_1$ for droplets with $D$ =20μm will be approximately 5ms, and 5μs, respectively; and for droplets with $D$=2mm, 0.5s and 5ms, respectively.

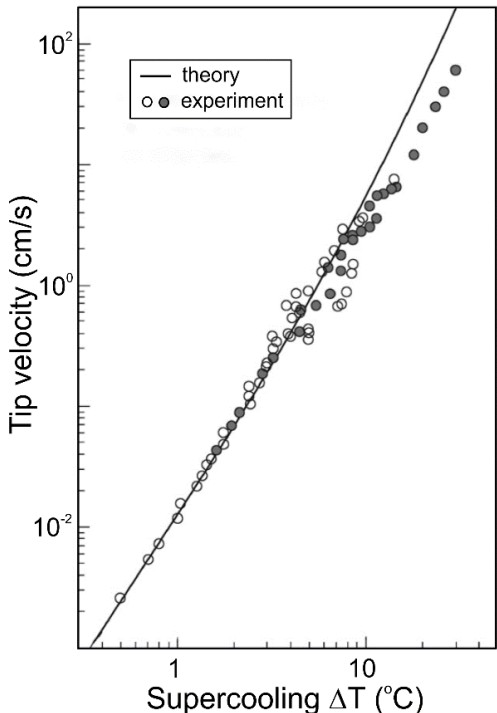

**Figure 3.** Summary of the measured velocity of freely growing ice as a function of supercooling measured by Lindenmeyer et al. (1959), Hallett (1964), Pruppacher (1967a), Kullinghall and Barduhn (1977), Furukawa and Shimada (1993), Feuillebois, et al. (1995); Ohsaka and Trinh (1998) (open circles), Shibkov et al. (2003) (solid circles). The
theoretical curve is based on Langer and Müller-Krumbhaar (1978) results. Adapted from Shibkov et al. (2003)

During the freezing stage droplets are cooling due to the thermal exchange with the ambient environment, and thus, the remaining liquid water gradually freezes. The second stage is quasi-isothermal and it is approximately 100-1000 times slower than the first stage. According to Pruppacher and Klett (1997), the time
of the second stage of the droplet freezing inward can be estimated as:

$$t_2 = \frac{\rho_w L_m D^2 (1 - \frac{\Delta T c_w}{L_m})}{12 f \Delta T \left( K_a + L_s D_v \left( \overline{\frac{d\rho_v}{dT}} \right)_{sat,i} \right)} \tag{4}$$

where $\rho_w$ liquid water density; $f$ the ventilation coefficient; $D_v$ is the water vapor diffusion coefficient; $K_a$ is the thermal conductivity of the air, $L_s$ latent heat of ice sublimation; $\left( \overline{\frac{d\rho_v}{dT}} \right)_{sat,i}$ is the mean slope of the ice saturation vapor density curve over the interval from $T_0$ to $T_m$. The ventilation coefficient $f$ describes the acceleration of droplet freezing from forced (due to the velocity between droplet and gas) and free (due to the temperature difference between droplet and gas) convection as compared to stagnant air ($f$=1). For drizzle sized droplets falling freely in the air, $f$ will typically assume values between 2 and 4.

Following Eq.4 at $T_a$ =-4°C and -20°C, the freezing time $t_2$ for droplets with $D$ =20μm will be approximately 70ms, and 11ms, and for droplets with $D$=2mm, 80s and 13s, respectively.

Since $t_2 \gg t_1$ the droplet freezing time is determined by the freezing during the second stage. Experimentally, the freezing time was studied by Muchnik and Rudko (1962), Murray and List (1972), Hindmarsh et al. (2003).

It should be noted that there is a good wealth of theoretical studies on the freezing time $t_2$ (e.g. Macklin and Payne, 1967; King, 1975; Gupta and Arora, 1992; Feuillebois et al., 1995; Tabakova et al., 2010). However, Eq.4 (Pruppacher and Klett, 1997) provides a reasonably accurate assessment of $t_2$, which is in good agreement with experimental measurements.

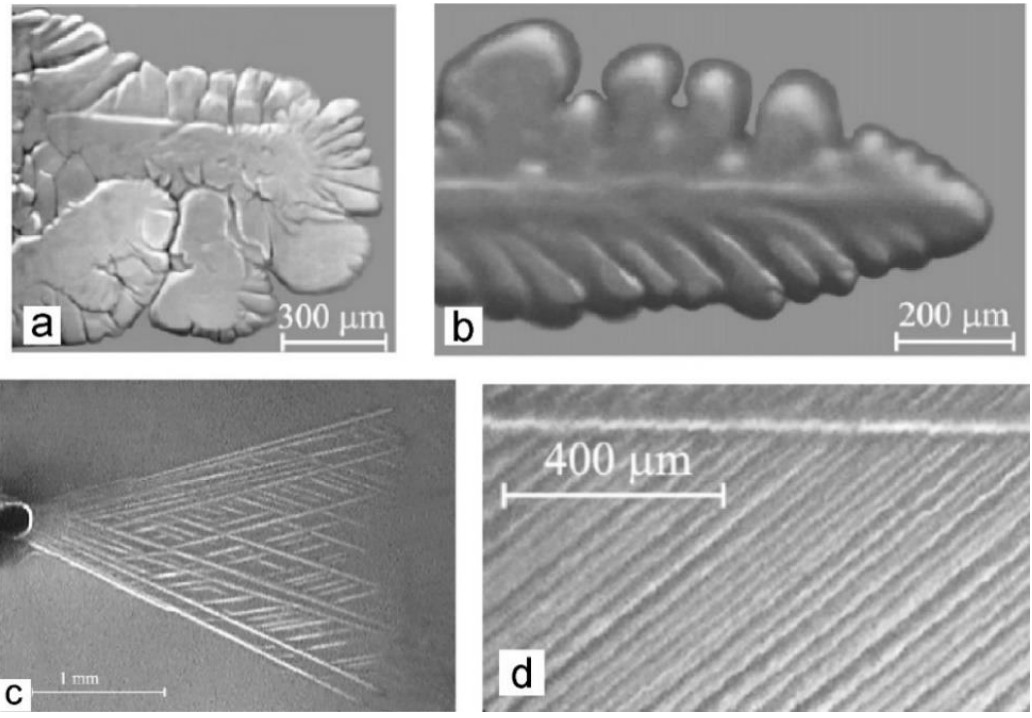

**Figure 4.** The morphology of ice crystal habits freely growing in pure water, supercooling at (a) $\Delta T$ =0.3°C dense brunching structure; (b) $\Delta T$ =1.5°C, developed dendrite; (c) $\Delta T$ =4.1°C, needle-like crystals; (d) $\Delta T$ =14.5°C compact needle mesh. (adapted from Shibkov et al. 2003)

**2.4 Crystalline structure of ice**

The way in which ice crystals grow through the freezing droplet during the recalescence stage is of great importance for SIP for two reasons. First it affects the formation of the ice shell and it also impacts the way in which the liquid water freezes inside the droplet. The morphology of ice formation during water freezing was explored by Kumai and Itagaki, (1953), Hallett (1960, 1964), Macklin and Ryan (1965, 1966), Pruppacher (1967a,b), Furukawa and Shimada (1993), Ohsaka and Trinh (1998), Shibkov et al. (2003, 2005). It was found that the shape of the ice crystals depends on the water supercooling $\Delta T$. At low supercooling (1°C< $\Delta T$ <3°C), ice crystals appear as stellar dendrites or dendritic sheets growing parallel to the basal plane. With the increase of supercooling, ice crystals start splitting, causing a formation of three-dimensional complex structures (e.g. Pruppacher 1967a; 1998; Shibkov et al, 2003). Splitting leads to so-called "non-rational" growth, i.e. growth that cannot be explained by rational crystallographic indices. Hallett (1964) and Macklin and Ryan (1965, 1966) suggested that this non-rational growth is explained by the hopper structure of ice crystal growth. One of the important findings of studies on water freezing is that the density of the ice mesh increases with the decrease of temperature, whereas the typical size of the ice crystals perpendicular to the *a*-axis becomes smaller. These transformations of ice crystals with temperature can be clearly seen in Fig.4. The shape of the ice crystals and the density of their network has direct impact on the size and the number of isolated water pockets formed during freezing as well as the tensile stress that is required to rupture the droplet.

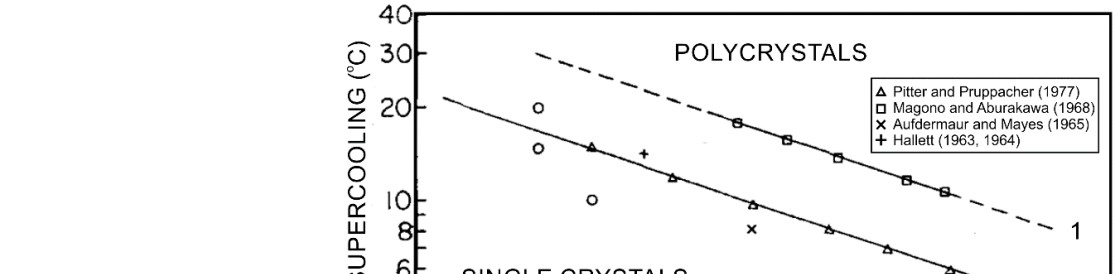

**Figure 5.** Dependence of the polycrystallinity of frozen droplets on the average droplet size frozen as single crystals and freezing temperature of droplets. (1) Droplets freely suspended in vertical airflow and nucleated by contact with clay particles. (2) Droplets frozen on the surface of large single crystals. Adapted from Pitter and Pruppacher (1973).

Regardless of the visual randomness of crystals growing through supercooled water, the non-rational structures may compose single crystals after the droplet freezing is completed (Macklin and Ryan 1965, 1966). Hallett (1963, 1964), Magono and Aburakawa (1969), Aufdermaur and Mays (1965), Pitter and Pruppacher (1973) studied the formation of monocrystalline and polycrystalline droplets during droplet freezing. They all found that droplet freezing as a single-crystal critically depends on the droplet size, supercooling of the droplet before freezing as well as the thermal conductivity of the medium, into

which the latent heat of freezing is dissipated. As it will be discussed below, droplet fragmentation during freezing and secondary ice production depend on whether droplets freeze as single-crystals or polycrystals.

The average critical radius of a droplet frozen as monocrystalline decreases with the increase of supercooling $\Delta T$ (Fig.5), and it can be parameterized as (Pitter and Pruppacher 1973):

$$r_c = \left(\frac{a}{\Delta T K^b}\right)^3 \tag{5}$$

where $K$ is the thermal conductivity of the medium surrounding the droplet; $a$=23; $b$=1/8. Eq.5 suggests that the lower the heat conductivity of the medium surrounding a given sized drop, the larger the supercooling can be reached when droplet freeze as monocrystal. Another important outcome from Eq.5 is that for the same $\Delta T$, due to lower value of $K$ for the air compared to ice, droplets nucleated by dust or miniscule ice crystals will have a larger monocrystal freezing size than cases where droplets freeze on the surface of a large ice particle.

### 2.5 Pressure inside freezing droplets

The pressure inside freezing drops was measured by Visagie (1969) and King and Fletcher (1973). Water drops with immersed pressure sensors were suspended in between paraffin oil and a carbon tetrachloride bath inside a temperature-controlled chamber. The size of the drops varied from 7mm to 12mm. It was found that, during freezing, the pressure inside a drop gradually built up as the shell became thicker. The pressure increase was repeatedly interrupted due to the complete or partial pressure relief brought on by cracking (Fig.6). In this period, water extruded through a crack and froze on the surface of the drop. After the crack was sealed by frozen water, the pressure would climb back to the previous value and continue to grow. Both studies showed that the pressure increased until reaching its maximum value $P_{max}$ near the point of complete freezing. The highest pressure, $P_{max}$= 89 bar in an 11mm diameter drop at -5°C, was observed by King and Fletcher (1973) and 79 bar in 7mm drop at -12.8°C by Visagie (1969). However, no appreciable pressure growth was observed inside drops freezing at temperatures higher than -3°C. King and Fletcher (1973) noted that about 20% of droplets contained a residual pressure of 10-20 bar at the completion of freezing.

The formation of cracks during droplet freezing was accompanied by an audible noise detected by microphone in the Visagie (1969) experiments. Loud sounds during droplet freezing and fragmentation were also reported by Dudetski and Sidorov (1911).

Visagie (1969) pointed out that besides the shell wall thickness, the cracking pressure is also a function of the temperature gradient across the ice shell (see Fig.6 in Visagie 1969).

King and Fletcher (1973) concluded that large droplet freezing at high temperatures will exhibit substantial viscous flow, and the smaller droplets freezing at lower temperatures will exhibit more elastic behaviour and crack more often. Between these two extremes, there is probably a size-temperature domain, in which sufficient elastic energy is stored in the shell to shatter it violently.

Both studies found that the cracking pressure increases with the increase of the thickness of the ice shell during the droplet freezing. However, the dependence of the cracking pressure versus droplet size and temperature remains unknown.

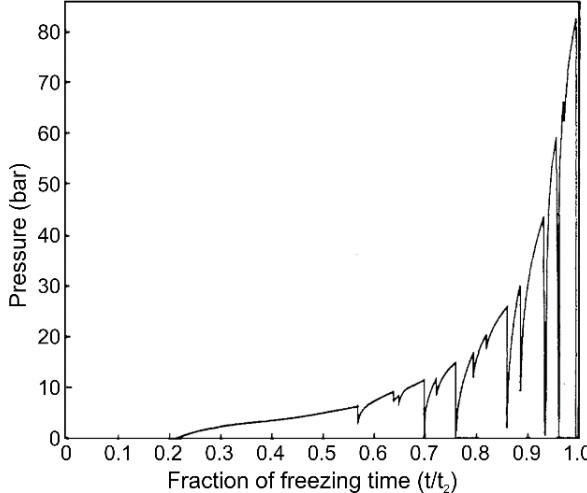

**Figure 6.** A time series of pressure changes inside an 11mm diameter drop freezing in a bath at -5C (adapted from King
and Fletcher, 1973).

     Visagie (1969) and King and Fletcher (1973) conducted their experiments with overly large drops (7mm to 11mm) placed in a paraffin oil and carbon tetrachloride bath. This experimental setup affects the temperature gradients in the ice shell around the freezing drops and the rate of heat exchange between the inner part of the
drops and their surrounding environment. These are the critical components for the cracking behavior and the inner pressure changes. This brings up the issue of whether the obtained results are applicable to drops of smaller sizes, which typically form in natural clouds.

### 2.6 Metamorphosis of droplet shape during freezing

     Visagie (1969) and King and Fletcher (1973) also documented that in addition to cracking, the release of
internal pressure inside freezing drops also occurred through deformation of the shape of the ice shell. Deformation of freezing drops was reported in early observations of freezing rain and ice pellets (e.g. Bentley, 1907). However, the physical explanations of freezing drop deformation were provided almost half a century later by Dorsey, (1948) and Blanchard (1951). Deformation of freezing droplets was observed by many authors in their laboratory studies (e.g. Mason and Maybank, 1960; Jonson and Hallett, 1968; Takahashi and
Yamashita, 1969; Pitter and Pruppacher, 1973; Takahashi, 1975, 1976; Iwabuchi and Magono, 1975; Pruppacher and Schlamp, 1975; Uyeda and Kikuchi, 1978; Lauber et al., 2018 and many others). Furthermore, Takahashi (1975) identified four main categories of drop deformation: (a) spike (Figs.7a and 8); (b) bulge (Fig.7bc); (c) split (Figs.7c), (d) crack (Figs.7d and 8). During freezing, droplets may simultaneously develop a combination different types of deformations depending on the droplet diameter and temperature, e.g. spikes
and cracks (Fig.8). Sketches of a variety of different forms of bulges, cracks and spikes are available from Takahashi (1975).

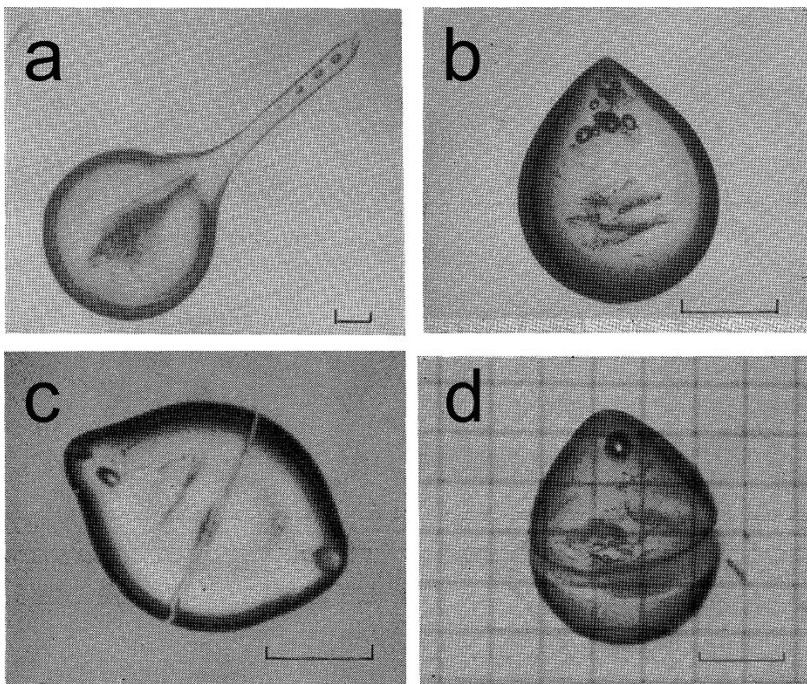

**Figure 7.** The main types of droplet deformations during freezing. (a) spike, long thin protrusion usually longer than one-fourth of the drop diameter; (b) bulge, protrusion shorter than one-fourth of the droplet diameter; (c) crack; (d) split. Scales are 100μm. Adapted from Takahashi (1975).

Takahashi (1976) found that deformation and shattering are closely related to crystalline structure formed during freezing. Thus, 90-100% of spikes are formed if droplets are polycrystalline. The spikes usually protrude from the crystal boundary whose mechanical connection is weaker compared to monocrystalline locations. Spikes are also formed if, at the moment of nucleation, the droplet temperature is higher than the ambient temperature. However, spikes scarcely formed when the droplet was in thermal equilibrium with the environment. Takahashi (1975) found that the probability of spike formation increases with the increase of droplet size. This can be explained by the increase of the occurrence of polycrystalline frozen drops with the increase of their sizes as in Fig. 5. Takahashi (1976) and Uyeda and Kikuchi (1978) studies also showed $c$-axis of a frozen monocrystalline droplet coincide with the $c$-axis of the seed crystal and that bulges are usually aligned with the $c$-axis.

Most experiments on observation of droplet deformation were performed with relatively large drops $D > 50$μm and at temperatures $T_a > -25°C$. However, López and Ávila (2012) observed the formation of spikes and bulges on small droplets with 8μm< $D_{eff}$ <30μm freezing at temperatures -40°C. Microphotographs of small frozen drops obtained in their experiments did not reveal cracks and splitting. The authors also did not find any evidence of shattering. However, no deformation of small droplets was observed by López and Ávila at $T=-30°C$. It is worth noting that the interpretation of López and Ávila is hindered by an absence of information about the nucleating temperature of droplets. Since the droplets were introduced in the cloud

chamber at positive temperatures, there is good reason to consider that they froze at temperatures higher than

        that of the environment. This kind of condition is favorable for spike formation (Takahashi, 1975). Deformed

        small droplets frozen at $T_a$<-40°C were also observed by Schaefer (1962).

        ### 2.7 Fragmentation during freezing

The following discussion will consider works focused on laboratory studies looking at the processes behind

        splintering and fragmentation of freezing droplets.

        Mason and Maybank (1960) studied the fragmentation of freezing droplets 30μm< $D$ <1mm in the

        temperature range of -25°C< $T_a$ <2°C. Droplets were suspended on a fiber in a small (~40cm$^3$) cloud

        chamber. It turned out that, on average, the occurrence of droplet shattering decreased with the decrease of air

temperature and droplet size. The occurrence of shattering for a 1mm diameter drop reached up to 47% with a

        maximum number of 200 splinters per drop. Such a high rate of splinter production may be an important factor

        in the INP economy during precipitation formation.

        However, Pruppacher (1967a) pointed out that when Mason and Maybank (1960) performed their

        experiments, droplets did not reach thermal equilibrium at the moment of nucleation, and their temperature $T_n$

was higher than $T_a$ by 1ºC to 12ºC. He argued that these conditions are favorable for the formation of an ice

        shell and for droplets freezing inward, which are critical for droplet shattering. Pruppacher questioned the

        relevance of the conditions used in the Mason and Maybank experiment to those in natural clouds.

        Dye and Hobbs (1968) and Johnson and Hallett (1968) attempted to reproduce the Mason and Maybank

        (1960) experiments. They found that a 1mm diameter water drop suspended on a fiber did not shatter when

nucleated after attaining thermal equilibrium. Dye and Hobbs (1968) also demonstrated that enhanced

        concentration of dissolved $CO_2$ resulted in increasing the occurrence of droplet shattering. They argued that the

        Mason and Maybank (1960) experiments were affected by increased concentrations of $CO_2$, which was used as

        a coolant. Johnson and Hallett (1968, Fig.2) also demonstrated that for the drops where the nucleating

        temperature was higher than that of the air ($T_n > T_a$), the ice shell forms around a pure liquid core and ice

mesh does not penetrate their centre. Such drops may create a stronger ice shell with a higher internal pressure,

        and therefore, be more susceptible to shattering.

        Hobbs and Alkezweeney (1968), Takahashi and Yamashita (1969, 1970), Bader et al. (1974), Pruppacher

        and Schlamp (1975) found that during freefall, droplets shatter after reaching a temperature quasi-equilibrium

        with the environment. It is important to note, that these results are in disagreement with those obtained by Dye

and Hobbs (1968) and Johnson and Hallett (1968).

        Despite the differences in experimental setups, most laboratory studies showed a general trend that large

        droplets are more susceptible to shattering during freezing than small ones (summarized in Lauber et al. 2018).

        However, Takahashi (1975) found that the relationship between the occurrence of shattering, droplet diameter

        and air temperature is more complex. He showed that in the air temperature range -20C<$T_a$<-7C, free falling

drops have the highest occurrence of shattering in the size range 75μm<$D$<135μm. Whereas at $T_a$=-25C, the

probability of droplet shattering nearly monotonically increases from 50μm to 500 μm. Takahashi (1975) also

found that at $T_a$=-4C, droplets with 50μm <$D$<200μm do not shatter. In this regard, it is worth mentioning that

Brownscombe and Thorndike (1968) observed a 9% occurrence of shattering in droplets with 50μm <$D$<90μm

at -5C, which is in agreement with Keinert et al. (2020) reporting a 15% occurrence of droplet breakup at -5°C,

which occurred only under free fall ventilation but not in stagnant air.

Laboratory studies also did not show a consistency for the lower threshold diameter for droplet

fragmentation. Adkins (1960) found no splintering for droplets $D$<10μm. Hobbs and Alkezweeney (1968)

observed no fragmentation of droplets 20μm <$D$<50μm. Johnson and Hallett (1968) reported no shattering

observed for droplets 5μm <$D$<38μm.  However, Mason and Maybank (1960, Table 1) observed droplet

shattering in the size range of 30μm<$D$<80 μm when the droplets were at thermal equilibrium. The

inconsistency of the latter result may be related to the enhanced concentration of $CO_2$ in the laboratory setup. It

is worth noting, that based on the theoretical analysis of the energy balance, Wildeman et al. (2017) concluded

that symmetrically freezing droplets smaller than 50μm in diameter cannot shatter.

Ambient air temperature has a significant effect on the occurrence of freezing drop shattering. Both

Takahashi and Yamashita (1970) and Lauber et al. (2018) found that the maximum rate of shattering is

observed between -10°C and -20°C for droplets ranging in size 85μm <$D$<350μm. This is generally consistent

with the results found by Brownscombe and Thorndike (1968) for droplets with 80μm <$D$<120μm, although

their temperature range was limited by -15°C<$T_a$<-5°C. However, for large drops with $D$ >500μm the

maximum occurrence of shattering was observed at $T_a$<-25°C (Takahashi, 1975). Hobbs and Alkezweeney

(1968) found that the rate of shattering of droplets 50μm <$D$<150μm does not depend on the temperature over

the range -32°C<$T_a$<-20°C. Whereas, in his experiments, Takahashi (1975, Fig.7) found a strong temperature

dependence of droplets shattering in this size range.

A review of the laboratory studies showed that the reported rate of shattering during droplet freezing varied

significantly. For example, Takahashi (1976) found that the maximum rate of shattering for free fall droplets

(200μm< $D$ <350μm) at -20°C< $T_a$ <-10°C was close to 40%. However, Lauber et al. (2018) showed that,

for droplets suspended in electro-dynamic balance (EDB), the maximum shattering rate for the same

temperature size range is close to 12%. This, however, increased notably to about 25% when the experiments

were conducted under terminal velocity ventilation (Keinert et al., 2020). Brownscombe and Thorndike (1968)

observed a 14% rate of shattering for free fall droplets with 80μm <$D$<120μm freezing at -15C.

A significant inconsistency of the efficiencies of ice splintering and their dependency on temperature and

droplet size obtained by different research groups is quite evident. This poses a key question about the

differences in experimental setups and the potential effects of other parameters. Already, Johnson and Hallett

(1968) have pointed out the importance of the effect of ventilation on droplet shattering. When a droplet with

*D*=500μm was suspended on a thread and ventilated at an equivalent to the free fall speed, no shattering was observed. However, when the droplet was rotated around an axis perpendicular to the airflow, shattering and cracking invariably occurred. This finding raised questions about the realism of the experiments that had a droplet suspended with a fixed orientation on a fiber or other mount. Under these conditions, the thermal exchange between the droplet and the ambient air is different compared to the free fall condition.

Pitter and Pruppacher (1973) demonstrated that a droplet suspended in the airflow begins to tumble and spin immediately after nucleation, thus providing a radially more symmetric heat loss. Drop spinning after nucleation was also reported by Dye and Hobbs (1968), Kolomeychuk et al. (1975) and Keinert et al., (2020). Initiation of tumbling and spinning after droplet nucleation can likely be explained by the asymmetrical shape and heterogeneous surface roughness that builds up quickly after freezing, thereby leading to a fluctuating torque being exerted by the terminal airflow.

Takahashi (1976) also revealed the importance of the crystalline nature of ice that forms inside freezing drops to their subsequent shattering. He found that 90-100% of shattering occurs when drops freeze as single crystals. Takahashi also showed that splitting occurs perpendicular to the *c*-axis dividing the drop in two equal parts (e.g. Fig.7c,d). The equatorial cracking and splitting of freezing drops was also reported by Wildeman et al. (2017, Fig.3a,b) and Lauber et al. (2018, Figs.5,6). Takahashi (1976) systematized how drops may shatter with respect to their crystalline boundaries. In most cases of polycrystalline drops, their fragmentation occurs along the crystal boundaries, where mechanical connectivity is weaker.

One of the first classifications of 'types of fragmentation' during drop freezing goes back to the work of Stott and Hutchinson (1965). They nucleated 0.9mm to 1.9mm diameter drops that were suspended on a fiber at -1°C and then froze them at the air temperature of -15°C. Even though this particular arrangement is not fully relevant to conditions in natural clouds, it helped identify the most common patterns of the drop fragmentation. The droplet fragmentation was classified as follows: (a) violent shattering with multiple pieces; (b) central breaks or splitting; (c) spicule breaks with liquid; (d) spicule breaks after solidification; (e) spicule bubble breaks; (f) cracks.

Wildeman et al. (2017) conducted experiments with millimeter sized drops freezing on a super-hydrophobic substrate. The high-speed videos documented explosive shattering of freezing drops, which generated a cascade of ice fragment sizes. One of the videos (V2) documented secondary shattering of one of the fragments formed after primary shattering. This suggests that during droplet freezing, liquid water may form several pockets across the droplet volume, rather than one big unfrozen volume in the central part. As discussed above, the connectivity of unfrozen pockets of water inside the ice shell is likely to be controlled by the type of the ice network formed inside the droplet and temperature exchange between the droplet and environment.

One of the caveats of the Wildemann et al. laboratory setup is that the experiments were performed at very low pressure (3.4 $10^{-3}$atm), and the droplets cooled much faster than they would cool in the atmosphere.

Johnson and Hallett (1968) showed that below 0.13atm, every drop in their experiments shattered violently. In
this way, the results are not directly applicable to environmental conditions.

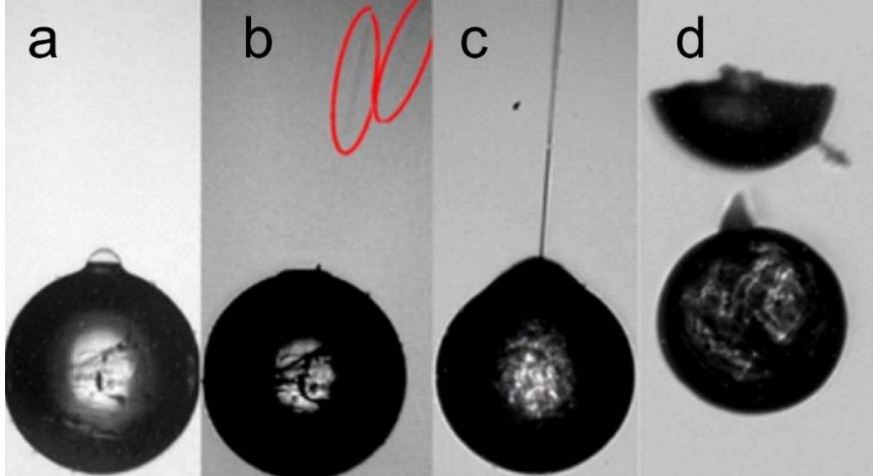

**Figure 8**: Secondary ice processes, as observed by high speed microscopy (a) a bubble has formed on the surface of a
freezing droplet. Cracks are visible in the surface. (b) the droplet from (a) but 12 ms later: the bubble has burst, two
fragments are highlighted by red ellipses. (c) jetting: a jet of liquid water is expelled violently through a hole in the ice
shell. (d) breakup: a freezing droplet splits in two halves, a few small fragments are sometimes observed (adapted from
Lauber et al. 2018).

In a series of experiments conducted with electrically charged droplets levitated in an electrodynamic

balance, Kiselev and colleagues observed droplet freezing with a high speed video microscope and categorized

secondary ice processes as breakup, cracking, bubble bursting and jetting, cf. Fig. 8. Opposite to previous

studies, they did not observe violent shattering of freezing droplets into many fragments. The relative and

absolute frequency of the secondary processes did not only depend on droplet size and temperature, but also on

droplet ventilation and the presence of solid inclusions or dissolved salts. The effect of solid inclusions

(polystyrene latex particles) was twofold. While they suppressed droplet shattering upon freezing of large

($D$=300µm) drizzle droplets (Lauber et al. 2018), they strongly enhanced droplet shattering in small ($D$=80µm)

drizzle droplets (Pander et al. 2015). Large droplets were found to shatter at higher temperatures and much

more frequently when suspended at terminal air velocity compared to being suspended in stagnant air under

otherwise identical conditions (Keinert et al. 2020). Dissolved sea salt hindered droplet shattering at all sizes at

concentrations above about 100 mg/L. It is reasoned that dissolved substances and solid inclusions are expelled

from the growing ice phase and concentrate in the liquid phase during freezing. Here, they hinder the

formation of a monocrystalline ice shell. So, on one hand, this reduces the pressure needed for breakup, but on

the other hand may open pathways for pressure release prior to breakup. Pressure release events such as jetting

or spiking have been found to occur. Once high concentrations of dissolved gases build up in the liquid phase

of the droplet interior, pressure release induces gas bubble formation in the droplet interior. These bubbles may

escape through spikes or cracks in the ice shell giving rise to bubbles. Upon freezing of the bubble skin, the

skin breaks and may form a source of additional tiny ice particles. Even though bubble bursting has been found

to be a frequent secondary ice process (Pander et al. 2015; Lauber et al. 2018), the number of emitted ice particles has not been quantified up to date. Droplet ventilation had a major influence on secondary ice process frequency and type (Keinert et al 2020). Droplets moving at terminal velocity with respect to the surrounding

air generally showed more frequent secondary ice processes when compared to droplets levitated in stagnant air. The dominant process observed shifted from cracking at stagnant conditions to breakup under free fall conditions. The latter could be observed even at temperatures higher than -5°C.

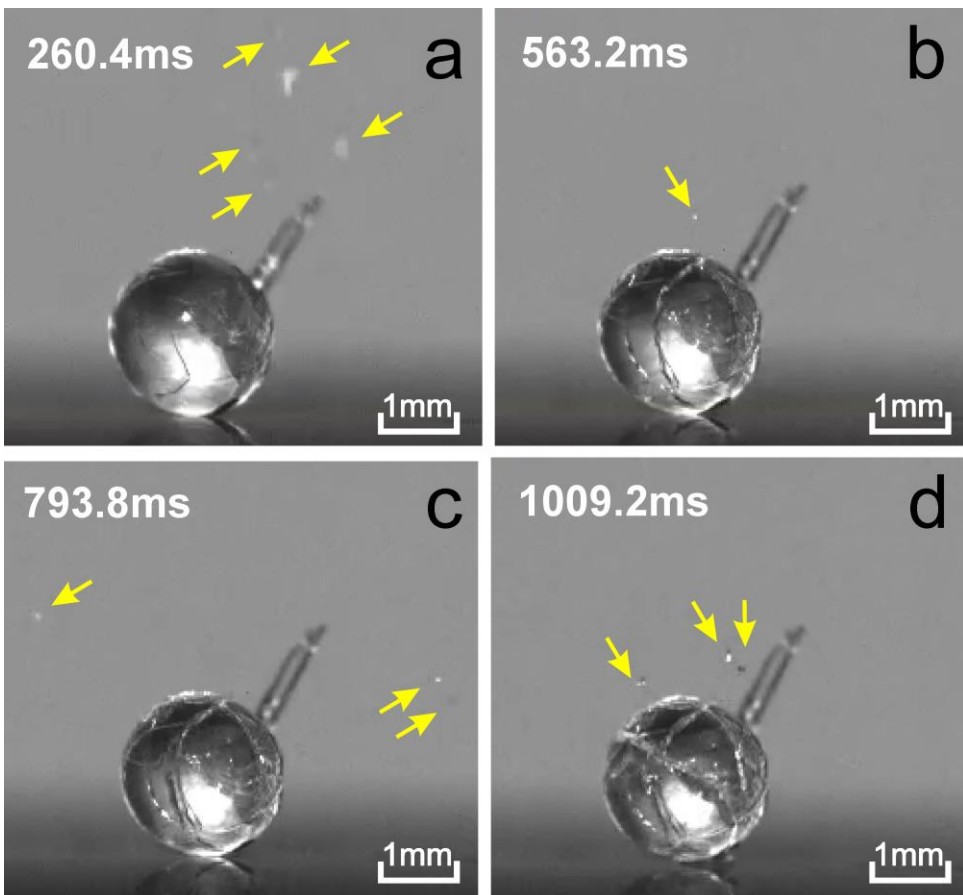

**Figure 9.** High-speed video snapshots of a 2mm drop at different stages of freezing. The pictures show progressive increase of the number of cracks covering the drop during its freezing. The yellow arrows indicate the locations of ice splinters ejected during cracking. The ambient temperature $T$=-7°C. The numbers in the top left corners indicate time since nucleation. Adapted from video V2 from the supplementary material to Wildeman et al. (2017).

King and Fletcher (1973) hypothesized that the numerous discontinuities in the pressure changes inside the freezing drops are indicative of the large-scale movements of the ice shell, and therefore, it may be a source of particles, even if the droplet does not shatter. This hypothesis was confirmed in experimental studies of droplet freezing by Wildeman et al. (2017). The production of ice splinters during cracking of 2mm freezing drop was documented in the supplementary high-speed video V2. Visual analysis of this video allowed for the

identification of several ice splintering events during cracking prior to final droplet shattering. Four of those events are shown in Fig.9. In general, the number of secondary ice particles due to droplet cracking during

freezing could be formulated as the product of the number of cracking events per freezing event and the average number of secondary particles per cracking event. However, the actual number of splintering during cracking events may be higher in comparison to those observed visually. This is because microphotography

allows for the detection of only those splinters that occurred within the depth-of-field of the microscope or whose sizes were larger than the detecting threshold of the optical system, and because not all cracking events are detectable by optical microscopy.

Splintering during cracking is an important finding, since it shows that freezing droplets may be a source of secondary ice even though they do not shatter by the end of freezing.


### 2.8 Summary

The review of the laboratory studies showed that the fragmentation of freezing drops is sensitive to a number parameters such as: (a) droplet size $D$, (b) environmental temperature $T_a$, (c) droplet nucleating temperature $T_n$, (d) air pressure $P$, (e) type of ice mesh formed during the recalescence stage (dependant on

$T_n$), (f) crystalline nature of freezing droplet (i.e. monocrystalline or polycrystalline), (g) thermal conductivity of surrounding medium $K$; (h) size of nucleating particle (small INP vs large ice particle, affects polycrystallinity); (i) ventilation $f(D, T_a, P)$ (e.g. static air, drop rotation during freezing, free fall), (j) fall velocity $u_z(D, T_a, P)$; (k) dissolved gases (specifically $CO_2$, dependent on $T_a$ and $P$). Several types of ice fragmentation during droplet freezing were documented: (1) splitting with few fragments, (2) explosive

shattering with multiple fragments, (3) cracking-splintering, (4) bubble bursting, (5) jetting. Unfortunately, the dependency of ice fragmentation during droplet freezing on the above parameters remains only partially understood.

A review of the laboratory studies on droplet freezing showed a large diversity of obtained results. The summary of the laboratory studies on droplet fragmentation during freezing is shown in Table 1. Thus, for a

single experimental setup under the same conditions, the number of fragments formed for the same size drop during its freezing varied from zero to a few hundred. Similarly, under the same laboratory conditions, studies observed that only a fraction of the droplets shattered, whereas the other fraction did not produce any fragments. This suggests that the laboratory experiments might contain hidden non-controlled parameters, which hindered obtaining reproducible results for each freezing droplet.

One of these parameters may be the orientation of the crystallographic axis of the INP with respect to the droplet surface at the moment of nucleation (Fig.10a,b,c). Since the growth rate of ice along the *a*- and *c*-axes is different (e.g. Macklin and Payne, 1968), the process of the droplet filling with the ice network during the recalescence stage may create different types of non-uniform temperature distribution inside the droplet, and ultimately affect the symmetry of the ice shell. In the case of a polycrystal INP it is expected that during the

recalescence stage a droplet will be filled by the ice network more uniformly (Fig.10c) as compared to a monocrystalline INP (Fig. 10ab).

Humidity of the surrounding environment may be another hidden aspect affecting SIP (Keinert et al 2020). Depending on the humidity level, the droplet may either grow or evaporate prior to nucleation. This may create additional temperature gradients at the droplet surface depending on its diameter. The near-surface temperature gradients may either hinder or facilitate the formation of the ice shell.


The topology of liquid volumes inside the freezing drop may also be an important factor for SIP. Thus, the cracking rate may be affected by the symmetry of the ice shell as well as the displacement of the liquid core with respect to the droplet center (Fig. 10d,e). The tensile stress formed in the ice shell is also expected to depend on how liquid water volumes are distributed across the freezing droplets: inside one big (Fig. 10d,e) or multiple small volumes (Fig. 10f). Unfortunately, no attention was given to this effect in previous laboratory studies.


There are a number of other parameters which received little attention in laboratory experiments that include: (a) size distribution of ice fragments, (b) minimum size of splinters, which may form during fragmentation, (c) minimal size for droplets to shatter, (d) effect of the angle between the *c*-axis and the droplet surface on ice shell formation, (e) humidity of the air.


Growing evidence from in-situ observations (e.g. Korolev et al. 2004, 2020; Rangno, 2008; Lawson et al. 2017) suggests that fragmentation during droplet freezing is an important SIP contributor to the concentration of cloud ice particles. Unfortunately, the diversity of laboratory results related to fragmentation during drop freezing hinders the development of a quantitative description and refined theory of this mechanism in order to use in cloud simulations. A variety of parameters and fragmentation types makes the experimental studies and quantification of this mechanism a challenging and intricate problem.


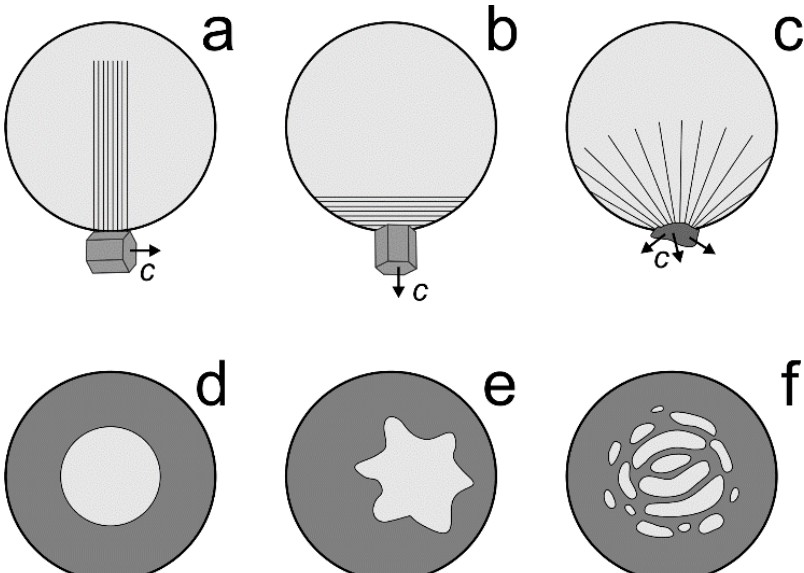

**Figure 10.** A conceptual diagram showing different possibilities of freezing of a supercooled droplet after nucleation by (a) monocrystalline INP or ice crystal with c-axis parallel to the droplet surface; (b) monocrystalline INP or ice crystal with c-axis perpendicular to the droplet surface; (c) polycrystalline INP or ice crystal. The visuals in (d, e, f) show various possible topologies of liquid zones formed during freezing: (d) idealized spherical liquid volume symmetrically centered with the ice shell (frequently used in numerical simulations of droplet freezing); (e) non-symmetrical liquid volume displaced towards the ice shell wall; (f) multiple disconnected liquid volumes.


**Table 1**. Summary of experimental studies of droplet fragmentation during freezing by different research groups. The table covers only works that quantified the parameters included in the table.

| Reference | Diameter (μm) | Temperature $T_a$ (°C) | Droplet suspension | Method of nucleation | Maximum SIP frequency (%) | max number fragments per drop | Temperature of maximum SIP rate |
|---|---|---|---|---|---|---|---|
| Mason and Maybank, 1960 | 30-1000 | -2 to -25 [1] | stagnant (fiber) | various [2] | 47 | 200 | -10C |
| Adkins, 1960 | 4-13 | n/a [3] | free fall | natural [4] | 0 | 0 | n/a |
| Hobbs and Alkezweeny, 1968 | 20-150 | -8 to -32 | free fall | various [5] | >5 | n/a | no temperature dependence |
| Brownscombe and Thorndike, 1968 | 50-90 | -5, -10, -15 | free fall | tiny ice crystals | 14 | 12 | -15°C |
| Dye and Hobbs, 1968 | 1000 | -3 to -15 | stagnant (fiber) | tiny ice crystals | 0 | 1 | no temperature dependence |
| Johnson and Hallett, 1968 | 1000 | -5 to -20 | stagnant (fiber) +ventilation | tiny ice crystals | >1 | n/a | no temperature dependence |
| Takahashi and Yamashita, 1969 | 600-800 | -18 to -25 | free fall | immersion [6] | 11 | n/a | -15°C |
| Takahashi and Yamashita, 1970 | 75-350 | 0 to -30 | free fall | tiny ice crystals | 37 | n/a | -15°C |
| Takahashi, 1975 | 45-765 | -4 to -24 | free fall | tiny ice crystals | 35 | n/a | -16°C |
| Pruppacher and Schlamp, 1975 | 410 | -7 to -23 | airflow | contact [7] | 15 | >3 | -11°C to -15°C |
| Bader et al., 1974 | 30, 42, 84 [8] | -10 to -30 | free fall | immersion [9] | n/a | 10 | n/a |
| Kolomeychuk et al., 1975 | 1600 | -12 to -25 | airflow [10] | natural [4] | 35 | 142 | -15°C to -18°C |
| Lauber et al., 2018 | 300 -320 | -5 to -30 | stagnant (EDB) | tiny ice crystals | 35 | 12 | -7C to -13C |
| Keinert et al., 2020 | 300- 320 | -1 to-30 | stagnant (EDB), airflow | tiny ice crystals | 1 | 3 | -10C to -15C |

1. ice nucleation temperature 0°C >$T_n$ >-15°C
2. natural nucleation, silver iodide, contact tiny small ice crystals
3. not available
4. no special efforts were made to nucleate droplets
5. natural or immersed silver iodide
6. kaolinite or silver iodide
7. kaolinite or montmorillonite
8. mean volume diameter
9. silver iodide
10. flow of humidified nitrogen

### 3. Splintering during riming

### 3.1 Efficiency of rime splintering

Splintering during ice particle riming is another mechanism that can explain SIP. Macklin (1960) observed splinter production in a small wind tunnel during the collection of droplets on an icing rod with 0.6cm diameter at temperatures -5C< $T_a$ <-20°C. The droplet diameters in their size distribution varied from few to 140μm (mean volume diameter~67μm) and their speed changed from 2m/s to 12m/s. A microscopic examination revealed long spicules and a few micrometer-sized ice features formed on the surface of the rod. The small fragile formations were hypothesized to be a source of the splinters. The ice crystal concentration during experiments was frequently observed to increase by a few orders of magnitude, reaching values of the order of $10^{-1}$cm$^{-3}$ at temperatures as high as -5°C.

Latham and Mason (1961) observed riming of freezing droplets on the hailstone simulator, accompanied by the ejection of ice splinters. They established that the splinter production varied with the air temperature, drop diameter and impact velocity. A maximum production rate of 14 splinters per droplet, was observed in droplets with diameter 70μm, impacting at 10 m/s at a temperature of -15°C.

Hobbs and Borrows (1966) and Aufrermaur and Jonson (1972) studied charge separation between an ice target and the flow of cloud particles on impact with each other. However, no significant ice splintering was found in both experiments. Hobbs and Borrows (1966) argued that the high rate of splintering observed by Latham and Mason (1961) may be related to carbon dioxide, which might be present in the experimental setup.

Bader et al. (1974) observed rime splintering during accretion of monodisperse droplets on a small copper target. The experiments were conducted for the ambient temperature ranging from -15°C<T<-9°C. Droplets fell at terminal velocity and deposited on the rimer. The number of accreted droplets per ejected ice splinters decreased from 2000 for droplets of 56μm diameter to 200 for droplets of 100μm diameter. These numbers correspond to 5 and 10 ice particles per milligram of rime, respectively. Ice particles were only ejected when there was an open chain-like structure on the rimer surface. However, no ice fragments were seen when water accretion was high enough to give a completely glazed deposit. The latter was likely associated with reaching the Ludlam limit (Ludlam, 1951).

Hallett and Mossop (1974) and Mossop and Hallett (1974) observed splinter formation during riming in a cloud chamber with liquid water content ~1g/m$^3$ and droplet concentration 500cm$^{-3}$. They found that splinter production is active in the temperature range -8°C< $T_a$ <-3°C. Furthermore, the rate of splinter production had a pronounced maximum at the air temperature of -5°C and the drop impact velocity 2.5m/s (Fig.11). At these conditions, one splinter was produced per 250 droplets of diameter *D*>24μm. The phenomenon of splinter production during riming is usually referred to as the Hallett-Mossop (HM) mechanism.

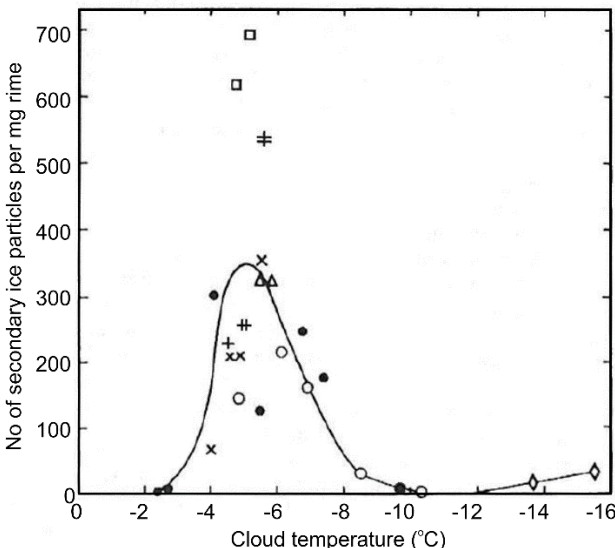

**Figure 11.** The dependence of the number of splinters per mg of rime on ambient temperature at speed 2.7m/s obtained experimentally by Hallett and Mossop (1974).

Mossop (1978, 1985) found that the presence of droplets with $D<12\mu m$ in addition to those $D>24\mu m$ increases the splinter production further. Saunders and Hosseini (2001) studied the splinter production in a wider range of impact velocities of up to 12m/s. They found that the maximum secondary ice ejection occurs at 6m/s with the number of splinters nearly five times lower than it was found in the Hallett and Mossop (1974) and Mossop and Hallett (1974) experiments.

The amount of dissolved gases is typically not specified in most laboratory experiments. Even if equilibrium has been reached, in nature this amount might depend on the chemical composition of the droplet, e.g. cloud droplet pH value.

Heymsfield and Mossop (1984) studied the effect of the rimer surface temperature on the production of secondary ice particles. They found that raising the surface temperature of the riming particle by 1°C transposes the splinter production curve virtually unchanged to air temperatures 1°C lower. This led them to conclude that splinter production due to the HM-mechanism may occur at air temperatures lower than -8°C depending on liquid water content (LWC) and the rimer fall velocity, which are the main factors determining the surface temperature of the riming particle. This conclusion is consistent with earlier work by Foster and Hallett (1982).

The quantification of the rime splintering production obtained from the experimental studies of Hallett and Mossop created a basis for various formulations of SIP parameterizations (e.g. Cotton et al. 1986; Meyers et al. 1997; Reisner et al, 1998 and others), which are widely used in numerical simulations of clouds.

### 3.2 Physical mechanism of rime splintering

Several studies are aimed at understanding the physical mechanisms responsible for splinter production. For instance, Macklin (1960) documented that fine ice structures formed during riming could be easily detached from the rimer and form splinters. One of these fine ice features are shown in Fig.12a.

Mossop (1976) proposed four possible mechanisms responsible for the HM-process: (1) formation of ice shell around accreted droplets with its subsequent fragmentation during freezing; (2) detachment of droplets that make glancing contact with rime; (3) growth and subsequent detachment of frail ice needles at temperatures around -5°C; (4) detachment of rimed ice by evaporation (see section 6).

Choularton et al. (1978, 1980) suggested that, if droplets $D$ >25µm are accreted to the ice substrate by a
thin neck, they will minimize the heat transfer toward the rimer. This arrangement may induce symmetrical heat loss to the air, which then leads to the formation of a complete ice shell around a droplet as it freezes. The freezing of liquid will result in a pressure build up inside the droplet which may cause shell disruptions with subsequent production of fragile protuberances of frozen water (Fig.12b). Mossop (1980) credited this hypothesis by pointing out that the ice shell is weakened by the presence of ammonia and results in a reduced
number of protuberances and splinters. He also showed that the increased ammonia concentration in droplets results in a reduction of the rate of splinter production. Griggs and Choularton (1983) suggested that the cut-off at about -8°C is due to rapid growth of the ice shell, which is too strong to be disrupted by the internal pressure.

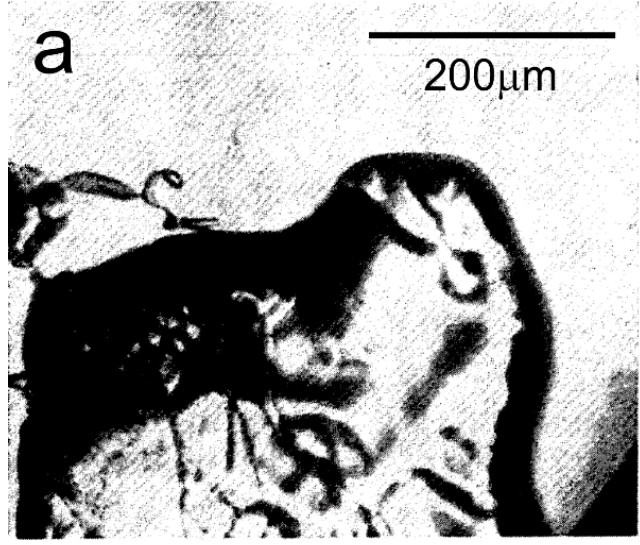 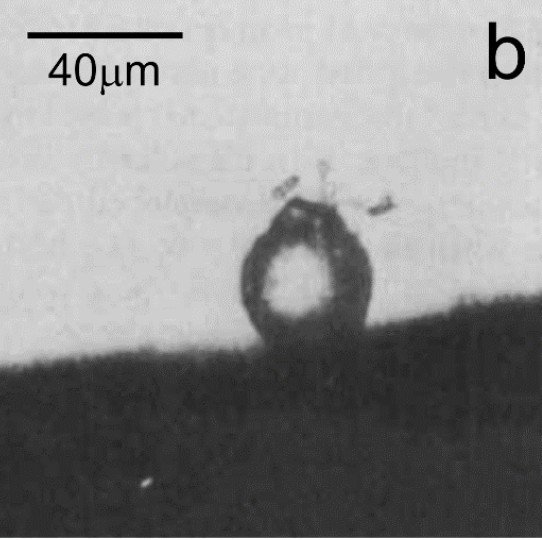

**Figure 12.** Pictures of the rime fragments:  (a) frozen splash observed at $T_a$ =-6°C and speed 6m/s (Macklin, 1960); (b) disruptions of protuberances formed out of a 35µm rimed droplet at $T_a$ =-7°C and speed 1.5m/s (Choularton et al. 1980)

Dong and Hallett (1989) reported that after impact with ice, droplets tend to spread over the surface of ice at all temperatures above -8°C. They concluded that splinter production by pressure build-up inside individual
frozen droplets is unlikely to be responsible for the shattering. They suggested that fragmentation is associated with the stress build up within an accreted droplet. This occurs when the droplet experiences a temperature gradient between the colder substrate and the surface of the droplet, freezing at 0°C. However, the absence of protuberances, when the droplets rime onto an ice surface above -10°C, contrasts with the observation of Choularton et al. (1980) who showed photographs of discrete frozen droplets together with protuberances
obtained in the temperature range -3°C to -7°C (Fig.12b).

Emersic and Connoly (2017) studied microscopic riming events on an ice target using a high-speed video recording. The droplet sizes ranged from 5 to 50 μm. It was found that the droplet behavior on impact depends on the uniformity of the rimer surface. Thus, droplets tend to spread flat on flat ice surfaces at temperatures associated with the HM-process, as was earlier observed by Macklin and Payne (1969) and Dong and Hallett (1989). However, with increasing rime depth, which is more commonly associated with graupel, growing rime spires protrude from the surface into the airflow around the rimer. No protuberances or liquid ejection was observed during the riming process, nor was any mechanical rime splintering event observed in approximately 1300 droplet freezing events. Based on the results of their study, Emersic and Connoly (2017) hypothesized that the rime spikes that develop with continuing droplet accretion could break off during particle tumbling or hinging by small droplets.

### 3.3 Summary

The literature review showed that, apart from some early studies (Hobbs and Borrows, 1966; Aufrermaur and Jonson, 1972), most laboratory experiments on the HM-process confirmed splinter production during riming. However, there was no consistency in the rate of the rime splintering observed by different groups. This can be clearly seen from Table 2 summarizing main laboratory results of the HM-process studies. This discrepancy is most likely related to different laboratory setups and techniques used for splinter counting.

The analysis of laboratory studies suggests that the efficiency of secondary ice production during the HM process depends on (a) air temperature $T_a$ (max efficiency at -8C< $T_a$< -3C); (b) surface temperature of the rimer $T_s$ (depends on $u_z$ and LWC); (c) size $L_r$ and density $\rho_r$ of the rimer; (d) fall speed of the rimer $u_z$ (determined by the size and density of the rimer and air density $u_z(L_r, \rho_r, \rho_a)$); (e) droplet size distribution $F(D)$. The condition applied to $F(D)$ requires presence of droplets with $D$ >24μm and $D$ <12μm. Another condition for $F(D)$ limits maximum LWC by the Ludlam limit (Ludlam 1951) when the rimer growth will turn into wet growth. In this case, the rimer's surface will be covered by a layer of liquid water, which will supress splintering.

To conclude this section, it should be emphasized that after several decades of rime splintering studies, the physical mechanisms behind this phenomenon are still under debate. Without clarifying the nature of this process, a development of a physically based parameterizations for numerical simulations of clouds does not seem to be feasible.

**Table 2**. Summary of experimental studies of rime splintering (HM-process). The table covers only works that quantified the parameters included in the table.

| Reference | Diameter (μm) | Temperature $T_a$ (°C) | Velocity (m/s) | Maximum rate of SIP (#splinetrs/mg) | Temperature of maximum SIP rate (°C) | Velocity of maximum SIP rate (m/s) |
|---|---|---|---|---|---|---|
| Latham and Mason, 1961 | 20-120 | -2 to -18 | 0 to 30 | ~8000 [1] | -8 to -18 | 5 to 15 |
| Hobbs and Borrows, 1966 | 30-70 | -4, -8 | 4 to11 | 0 | n/a [2] | n/a |
| Aufrermaur and Jonson, 1972 | 20-100 | -5 to -15 | 10 | 0 | n/a | n/a |
| Bader et al., 1974 | 56-100 | -9 to -15 | free fall | 10 | n/a | n/a |
| Hallett and Mossop, 1974 Mossop and Hallett, 1974 | <35 | -2 to -16 | 0.7 to 3.1 | 700 | -5 | 1.5 |
| Mossop et al., 1974 | 15+50 | -8, -10 | 0.8 | 0.24 | -8 | n/a |
| Mossop, 1976 | 3-45 | -5 | 1.4 to 3 | <550 [3] | n/a | 1.4 to 3 |
| Heymsfield and Mossop, 1984 | 2-40 | -2 to -9 | 1.8 | 220 | -4 to -6 [4] | n/a |
| Mosop, 1985b | 5-40 | -2 to -8 | 0.55 to 5 | 300 | -4.3 | 2 to 4 |
| Saunders and Hosseini, 2001 | 5-40 | -5 [5] | 1.5 to 12 | 70 | n/a | 6 |

1. 14 splinters per droplet 70μm diameter
2. not available
3. 1 splinter per 250 droplets >24μm diameter
4. depending on LWC
5. rimer surface temperature $T_s$

### 4. Fragmentation due to ice-ice collision

Collision of ice particles may result in their mechanical fragmentation and production of secondary ice (Langmuir, 1948, p.186). This hypothesis was stimulated by observations of ice particle fragments collected during airborne (e.g. Hobbs and Farber, 1972; Takahashi, 1993) or ground-based (Jiusto and Weickmann, 1973) studies.

There were only two known laboratory works on collisional ice fragmentation. Vardiman (1978) explored fragmentation of natural cloud ice particles on impact with a metal mesh. He found that "graupel is surprisingly ineffective in generating fragments". However, light to moderate rimed spatial crystals are the most efficient source of ice fragments. For planar crystals, the degree of fragmentation increases with the degree of riming.

Takahashi et al. (1995) studied the dependence of mechanical fragmentation resulting from collision of 2cm in diameter rimed ice spheres. The ice spheres were attached to the edges of 10cm long spinning metal rods and were made to collide with each other at a speed of 4m/s. This speed was used to simulate a fall speed of a 4mm diameter lump graupel with a density of 0.3-0.4g/m$^3$. The collisional force changed incrementally from 20dyn to 500dyn. Takahashi et al. found that the number of fragments depends on the degree of riming, temperature, and collision force. The maximum number of fragments per collision (up to 800) was observed at -16°C.

It is hard to judge the consistency of the results obtained by Vardiman (1978) and Takahashi et al. (1995) because of the differences in the experimental setups and environmental conditions. It is also difficult to identify the degree of applicability of the rate of SIP obtained in these experiments to free falling ice particles grown in natural clouds.

Collisional ice fragmentation was also studied theoretically by Hobbs and Farber (1972), Vardiman (1978), Phillips et al. (2017). These studies were based on the consideration of collisional kinetic energy and linear momentum. Such considerations would be relevant only for cases of direct central impact. In a general case, angular momentum and rotational energy should be taken into consideration. Since oblique particle collisions are more frequent than central collision, the efficiency of SIP obtained in these works is expected to be overestimated.

The theoretical considerations of collisional fragmentation in Yano and Phillips, (2011), Yano et al. (2016) and Phillips et al. (2017) were based on the rate of ice production from Takahashi et al. (1995). A detailed analysis of the Takahashi et al (1995) laboratory setup indicated that the riming of ice spheres occurred in still air, which resulted in more lumpy and fragile rime compared to that formed in free-falling graupel. The collisional kinetic energy and the surface area of collision of the 2cm diameter ice spheres also significantly exceed the kinetic energy and collision area of a few mm sized graupel. Altogether, it may result in overestimation of the rate of SIP, compared to graupel formed in natural clouds.

It should also be mentioned that ice particle fragments observed in-situ (e.g. Hobbs and Farber, 1972) may be a result of particle breakups induced by the sampling instrument (see section 8). Schwarzenboeck et al. (2009) identified that 18% of observed incomplete dendrites are the result of natural fragmentation. The identification of natural fragments was based on the observation of the ice shapes near the expected break area,

which were interpreted as "subsequent growth". However, it could be argued, that incomplete dendrites may naturally form because of growth suppression of one or more branches due to defects or dislocations on the crystal. Examples of 1-, 2-, 3- and 4-branched stellar and dendritic crystals with underdeveloped defected branches were documented by Bentley and Humphreys (1962, pl.198-204), Auer (1970, Figs.9,28,30), Kikuchi and Uyeda (1978, Fig.2).

The analysis of literature suggests that the efficiency of SIP during ice-ice collision depends on (a) properties of the colliding particles, such as size, mass, density, shape, surface roughness; (b) air temperature $T_a$ (determines physical properties of ice, e.g. crispness), (c) relative fall velocity of colliding particles (depends on the aerodynamic size of particles and air density).

In summary of this section, it can be concluded that the efficiency of SIP during ice-ice collisional

fragmentation remains uncertain due to the lack of laboratory studies. No parameterizations of SIP due to ice-ice collisional fragmentation can be developed at that stage based on two laboratory observations, whose results are conflicting with each other. Additional laboratory studies are required to explore ice-ice collisional fragmentation of free-falling ice particles with different habits. Ice fragments observed in-situ should be considered with caution due to potential particle breakups during sampling (see section 8).

## 5.   Fragmentation due to thermal shock

When a supercooled drop rimes on the surface of an ice crystal, it freezes, and its temperature rises to the melting point (section 2). Some fraction of the latent heat released during freezing will be transferred into the ice crystal. Koenig (1963, p.35) hypothesized that this may cause a thermal shock at the location of the droplet

attachments with following ice crystal cracking and splintering due differential expansion of ice.

From his lab experiments, Gold (1963) found that the surface temperature shock of 6°C is necessary to produce the stress required for ice cracking.

Dye and Hobbs (1968) observed during laboratory experiments that, when an ice crystal on some occasions became attached to a freezing drop, it would often break into 5 to 10 pieces as the drop froze. Sometimes, the

breakup of the crystal would occur when the drop cracked. On other occasions the crystal would break without any apparent changes to the freezing drop. Later Hobbs and Farber (1972) reproduced laboratory experiments of Dye and Hobbs. They observed shattering of a dendritic crystal into several pieces after bringing it in contact with 2mm diameter supercooled drop. These observations are of considerable interest, for it suggests that the breaking up of ice crystals that collide and nucleate supercooled drops, may play an important role in

increasing the concentration of ice particles in natural clouds.

Using thermoelastic theory, King and Fletcher (1976a) calculated thermal stresses in idealised ice shapes on impact with liquid droplets, when a small area was warmed to 0°C. They concluded that a thermal shock mechanism is unlikely to be responsible for SIP at temperatures $T_a$>-5°C.

King and Fletcher (1976b) conducted a series of experiments to study the effect of thermal shock on cracking of macroscopic polycrystalline spheres ($D\sim$2-3cm) and thick ($\Delta h$=1.7cm) and thin ($\Delta h$=1-2mm) cylindrical plates with diameter $b$=5cm at temperatures down to -40°C. The cracking probability of ice plates versus temperature was studied for several ratios $a/b$=0.2, 0.4 and 0.6, where $a$ is the diameter of the heated area. In such arrangement the thermal shock is expected to be more severe than that experienced by microscopic ice crystals during riming. Depending on the thickness of the plates and the ratio $a/b$, the cracking temperature threshold varied from -5°C to -35°C. None of the plates fragmented or separated. King and Fletcher concluded that thermal shock is unlikely to be an important ice multiplication mechanism at -5°C.

Experiments using thermal shock with macroscopic slabs and spheres by King and Fletcher (1976b) are not fully scalable down to microscopic monocrystalline ice particles. Moreover, the conclusions obtained in their studies are not consistent with the laboratory observations of Gold (1963), Dye and Hobbs (1968) and Hobbs and Farber (1972).

The review of the studies of SIP due to fragmentation of ice particles due to thermal shock depends on (1) air temperature $T_a$; (b) mass and shape of ice particle; (c) droplet diameter; (d) local geometrical configuration of ice particle at the location of the droplet attachment; (e) relative velocity of droplet and ice particle at the moment of impact (determines thermal connection of the droplet and ice particle); (f) ventilation (determines convective heat losses to the air).

Despite the seeming feasibility of this SIP process to occur in natural clouds, this phenomenon got little attention from the cloud physics experimental community. Based on the previous experimental and theoretical studies, the efficiency of ice fragmentation due to thermal shock is expected to primarily depend on the air temperature, droplet size, ice crystal size and its habit. Unfortunately, none of these dependencies have been addressed experimentally. Therefore, the effect of thermal shock ice fragmentation on SIP remains inconclusive.

## 6. Fragmentation of sublimating ice particles

Ice particle fragmentation and formation of secondary ice may occur during sublimation in subsaturated cloud regions. Mossop et al. (1974, section 4c) observed sublimation of rimed particles under the microscope. They reported detachment of 20μm size rime from their original locations and identified it as an "unexpected phenomenon", which may explain SIP.

Oraltay and Hallet (1989) studied evaporation of ice particles suspended on a fiber at a wind speed emulating their fall velocity. They observed the fragmentation of dendritic ice shapes at subfreezing temperatures only when relative humidity over ice was $RH_i$<70% and ventilation velocity was 10 to 20cm/s.

However, no sublimation breakup was observed for columnar and plate-like crystals. Dong et al. (1994) studied fragmentation of rimed ice and needles at 50%<$RH_i$<90%, -18ºC<$T_a$<-5ºC and ventilation speed ~1m/s. In their experiments, they found that a few mm long rimed ice particles may generate up to 100 fragments during evaporation at $RH_i$<70% within 1-2 minutes.

Bacon et al. (1998) studied fragmentation of sublimating ice particles suspended in electrodynamic balance inside a thermo-diffusional chamber at 85%<$RH_i$<100% and -30C<$T_a$<0ºC. The observed fragmentation tended to affect prolate ice particles with an aspect ratio higher than 3. An example of images of sublimating ice particle is shown in Fig.13. All three studies concluded that breakup rates depend on temperature and humidity, but largely on the initial shape of the ice particle.

During in-situ observation of metamorphosis of shapes of sublimating ice particles in natural clouds, Korolev and Isaac (2004) came to a conclusion that ice particle fragmentation during sublimation does not play an important role in SIP.

The laboratory experiments suggest that fragmentation of ice particles depends on (a) ice particle shape and size; (b) relative humidity; (c) pressure; (d) air temperature; (e) air temperature $T_a$.; (f) fall velocity and

ventilation coefficient.

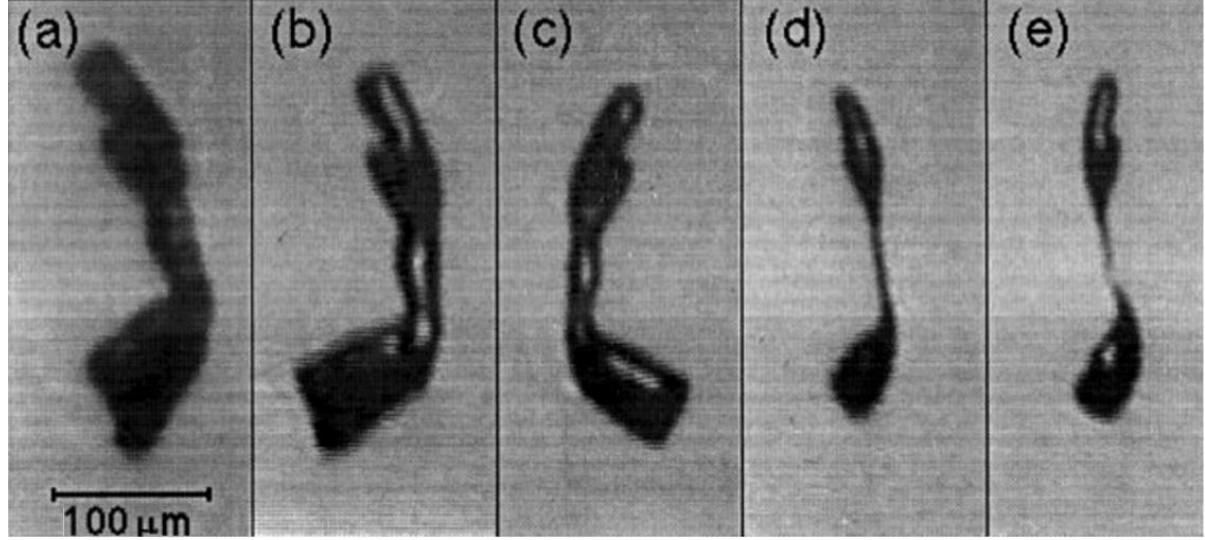

**Figure 13.** A sequence of images of a sublimating ice particle levitated in an EDB. Time before breakup (a) 6 min, (b) 4 min, (c) 2 min, (d) 20 s, and (e) at the moment of breakup (Bacon et al. 1998).

In order for the ice fragments formed during sublimation to result in ice multiplication, they have to re-enter back into a supersaturated cloud region. Since small ice fragments have lower terminal fall velocity, their residence time in the undersaturated environment may be long enough to result in their complete evaporation before they can re-enter a supersaturated environment. This appears to be a significant limitation of the SIP mechanism due to sublimation breakup. This mechanism is also unlikely to explain explosive concentrations

of small ice crystals frequently observed in convective and stratiform frontal clouds (e.g. Lawson et al., 2017; Korolev et al, 2020).

## 7.    Activation of INPs in transient supersaturation around freezing drops

Muchnik and Rudko (1961) and Dye and Hobbs (1968) reported observation of a halo of small droplets formed around a freezing drop immediately after the moment of its nucleation. Dye and Hobbs (1968) explained the origin of small droplets by the activation of CCNs in the region of high transient[1] supersaturation formed around freezing droplets. After ice nucleation, the droplet surface temperature $T_s$ rises to 0°C. Under the condition that the surrounding air has $T_a$<0°C, the surface of the freezing drop acts as a source of water vapor to a colder environment. The resulting water vapor diffuses radially outward. Depending on the air humidity, it may create at some distance from the droplet a region with supersaturated air. Nix and Fukuta (1974) developed a theoretical framework for the calculation of the supersaturation field around a stationary freezing drop, which was determined by molecular diffusion. They showed that maximum supersaturation increases with the decrease of $T_a$ and the increase of drop size.

Cheng (1970) attempted to explain the origin of small droplets due to their ejection from the freezing drop. However, this explanation was challenged by Hobbs (1971). Rosinski et al. (1972) also described laboratory results refuting Cheng's interpretation of the halo around freezing drops.

Later, Gagin (1972) proposed a mechanism explaining SIP due to activation of INP in high transient supersaturation area around freezing drops. He argued that high supersaturation may result in activation of insoluble INPs, which normally do not activate at typical cloud supersaturation levels ($SS$ <1%).

Rosinski et al. (1975) studied activation of silver iodide and soil particles placed on a flat plate at different distances from 2mm freezing drops. They found that silver iodide nucleated as water at temperatures $T_a$>-9.8°C, and as ice at $T_a$ <-9.8°C. Soil particles with sizes 20µm and 40µm nucleated as water at temperatures -20°C and -16°C, and as ice at lower temperatures, respectively. Rosinski et al. concluded that "production of ice particles by condensation-followed-by-freezing in a parcel of a cloud containing large freezing drops is orders of magnitude higher than by contact nucleation".

Gagin and Nozyce (1984) suspended 1mm-2mm diameter drops inside a gradually cooling chamber. The drops froze at a mean temperature -6.5°C as they contained silver iodide. Complete drop freezing occurred in 5-6 minutes, when the ambient temperature decreased down to -10°C -12°C. From the aerosol in the ambient air they found that during drop freezing on average 1.6-2.1 ice crystals were activated around freezing drops. The nucleation of ice crystals was attributed to supersaturation sensitive INPs.

In laboratory experiments of Rosinski et al. (1975) and Gagin and Nozyce (1984) the transient supersaturation in addition to the molecular diffusion was also contributed by a mixing with a convective flow, induced by the temperature difference between the drop surface and environment. None of the above studies accounted for a ventilation effect for free falling drops. In this regard, it was not clear whether the obtained

---

[1] Some studies use the term "transitional".

results are applicable to natural clouds, since the mixing will occur in the wake of falling drops, and it will be

mainly determined by turbulent mixing.

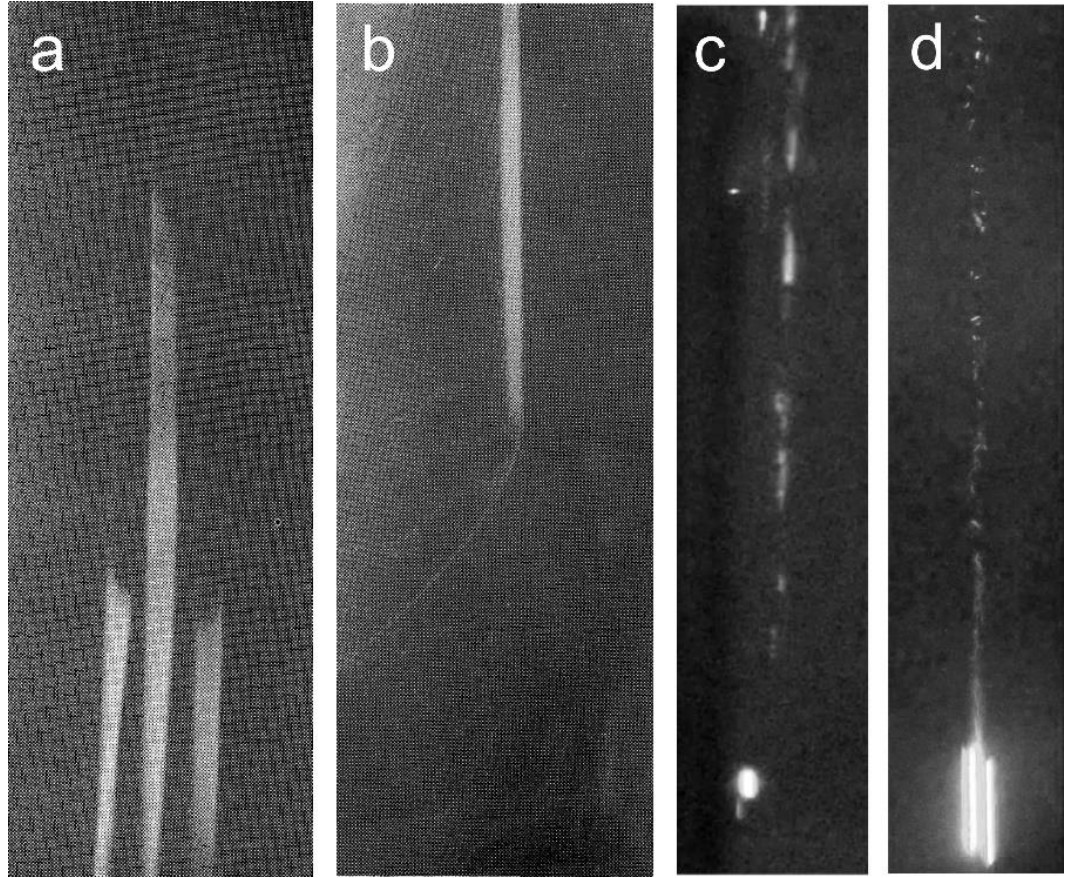

**Figure 14.** The trajectories of falling drops with $D = 100\mu m$ visualised by nucleation of CCN and INP in the high supersaturation formed around freezing drops at the moment of (a) initiation of freezing and (b) by the end of freezing at
ambient temperature $T_a \approx -65°C$ (Iwabuchi and Magono, 1975). Nucleation of water droplets and ice particles in the wake of falling with $D \approx 2mm$ (c) $T_a \approx -20°C$, $RH_w \approx 60\%$, $0C < T_a < 2°C$; (d) $T_a = 18°C$, $RH_w = 60\%$, $T_s = 10°C$. In these experiments AgI and of Snomax were used as ice nuclei (Prabhakaran et al., 2020).

Iwabuchi and Magono (1975) in their experiments on freezing electrification documented formation of fog
along the trajectories of 90$\mu$m-160$\mu$m free falling freezing drops at $T_a = -65°C$ (Fig.14ab). They used the fog

formation to identify the start and end moments of the drop freezing.

Prabhakaran et al. (2020) conducted experiments with a 2mm diameter free falling drops in an environment

with $T_a = -18°C$ and $RH_w = 60\%-80\%$. The experiments were conducted using Snomax and AgI aerosols

induced into the ambient air. The drop temperatures varied in the range $0°C < T_s < +20°C$. The high drop
temperatures were used to enhance supersaturation and exaggerate ice nucleation in the undersaturated air. The

free-falling drops formed fog trails consisting of activated cloud droplets and ice (Fig.14cd).

Nix and Fukuta (1974) also pointed out that hailstones during wet growth have a surface temperature close

to 0°C, and therefore they may act as a source of high supersaturation. Under such conditions, hailstones may

activate many more supersaturation sensitive INPs than a freezing droplet, since the affected volume in such a

case will be much larger. Fukuta and Lee (1986) performed calculations of supersaturation around falling

graupel with different sizes (2mm, 4mm, 6mm) at different ambient temperatures (-10°C, -20°C, and -30°C).

They found that larger graupel with larger sweeping volume has lower maximum supersaturation. Thus, over

2mm and 6mm falling graupel maximum supersaturation with respect to water at -10°C, -20°C and -30°C

reaches approximately 10%, 40%, 100% and 5.5%, 23%, 35%, respectively. The finding that for falling

freezing drops maximum supersaturation is decreasing with the increase of the drop size is opposite to that for

stationary drops in Nix and Fukuta (1974).

Chouippe et al. (2019) performed direct numerical simulations (DNS) of a free-falling ice sphere in humid

air accounting for heat and mass transfer. This study was focused on exploring accuracy of numerical

simulation. It confirmed the conclusion obtained in previous studies that supersaturation increases with the

increase of the temperature difference $\Delta T = T_s - T_a$. Krayer et al. (2020) used the same DNS model. They

found that significant values of supersaturation can be attained in the wake of warm hydrometeors, which

persist long enough to be observed at more than 50 particle diameters downstream of the meteor for

sufficiently high differences in temperature. The supersaturated volume of air exceeds the estimations by

Fukuta and Lee (1986) by far, which is attributed to the more accurate representation of the flow in the DNS

model.

It is worth noting, that high transient supersaturation may form not only over a particle with a surface

temperature $T_s > T_a$. Similar effect may also occur over a graupel or hailstone, for which the surface

temperature did not reach equilibrium, and remains lower than the ambient air. Thus, Schaefer and Cheng

(1971, Fig.1a) observed initiation of ice around a simulated graupel with temperature lower 5°C than the

ambient air temperature. Unfortunately, no other details of the experimental setup were available from their

work.

The above studies suggest that the activation of INP is expected to grow with the increase of the

temperature difference $T_s - T_a$. However, Baker (1991) argued that even if $T_a$ is as low as -15°C, the total

volume with high supersaturation around all freezing drops remains too small to enhance the number

concentration of active INP by several orders of magnitude. Therefore, INP activation in transient

supersaturation around freezing drops should have a low significance for SIP. This result seems to be

conflicting with the conclusion obtained in Rosinski et al. (1975) and Prabhakaran et al. (2020). It should be

noted that Baker (1991) assessment of the SIP efficiency was obtained for the static field of supersaturation

around droplet and under the assumption, that the number concentration of active INP follows a power law in

supersaturation that could be extrapolated to a very high supersaturation.

In summary of this section it can be concluded that efficiency of SIP due to INP activation in transient

supersaturation depends on (a) droplet diameter $D$; (b) air temperature $T_a$; (c) droplet surface temperature $T_s$;

(d) droplet fall speed $u_z$; (e) relative humidity $RH$; (f) air density $\rho_a$; (g) turbulence intensity $\varepsilon$; (h) droplet

freezing time (depends on $D, T_a, \rho_a, u_z, RH, \varepsilon$); (i) concentration and nucleation activity of interstitial INPs.

The studies described above provide experimental and theoretical support that activation of supersaturation sensitive INPs in the wake of free-falling freezing drops, wet hailstones or riming graupel is one of possible mechanisms of SIP. Unfortunately, due to limited experimental studies, the effect of INP activation around falling hydrometeors cannot be quantified and employed in cloud simulations. Future laboratory studies should be focused on the behaviour of INP at very high ice supersaturation (>10) for a better the quantification of the

effect of $T_s - T_a$ of a free-falling hydrometeor on the INP activation.

### 8.  Spurious enhancement of ice concentration during sampling

   In this section we discuss results of experimental studies of artificial fragmentation of ice particles during in-situ sampling. Artificial ice particle fragmentation may result in a significant enhancement of the measured

ice concentration and be confused with SIP. Airborne in-situ measurements is the main source of information about the concentration of ice particles in natural clouds and the environmental conditions associated with SIP. The accuracy of in-situ measurements of small ice particles is of great importance for the closure of SIP parameterizations and provides feedback to laboratory studies.

   At the initial stage of regular cloud observations with optical particle probes (Knollenberg, 1981) it was

found that small ice particles were observed in all ice clouds including precipitating and undersaturated cloud regions where existence of small particles conflicted with their small fall velocity and rapid sublimation, respectively. Such observations required developing additional mechanisms to explain the omnipresence of small ice crystals.

   The hypothesis of enhanced ice concentration induced by airborne instruments has been discussed over a

long period of time. Larger ice particles may bounce off the forward probe's tips or inlet, and shatter into smaller fragments. After rebounding, the shattered fragments may travel into the probe's sample volume and cause multiple artificial counts of small ice. Cooper (1977) was the first to recognize a potential significance of instrumental particle shattering and suggested filtering the shattered artifacts based on the characteristically short interarrival times between successive particles passing through the probe's sample volume. Several

following works based on comparisons between several airborne instruments (Gardiner and Hallett, 1985; Gayet et al., 1996) or analysis of the particles' interarrival time (Field et al., 2003) posed the question of whether the observed high concentration of ice particles is real or an artifact.

   Korolev and Isaac (2005) documented OAP-2DC, OAP-2DP, and HVPS images of fragmented precipitation size ice particles as a direct evidence of the existence of shattering. However, it did not clarify the

origin of the enhanced concentration of small ice.

   Field et al. (2006) applied an interarrival time algorithm to identify and filter out shattering artifacts in OAP-2DC and CIP measurements. It was found that after filtering artifacts, the OAP-2DC and CIP concentrations were reduced by up to a factor of four, when the mass-weighted mean size exceeded 3mm.

Heymsfield (2007), McFarquhar et al. (2007), Jensen et al. (2009), Vidaurre and Hallett (2009) based on
the comparisons between different airborne instruments, built up more evidence about the spurious enhancement of concentration of small ice particles.

Despite the growing evidence of the significance of the effect of shattering on ice particle measurements, the shattering hypothesis was not commonly accepted in the cloud physics community for many years. Many researchers argued that shattered particle fragments, after bouncing from the probe's upstream surface, shed
along the surface of the arms or inlets, and that they could not travel several centimeters across the airflow at an aircraft speed of 100m/s to reach the probe's sample volume.

A direct experimental support for the shattering hypothesis has been provided by a series of wind tunnel experiments with controlled environment and reproducible ice spray conditions (Korolev et al. 2011, 2013b). Ice particle impacting with the probe tips at aircraft speed were video-recorded by a high-speed camera. These
videos documented that after rebounding from the probe's tips, shattered small fragments can travel several centimeters across the airflow and reach the probe's sample volume (Fig.15).

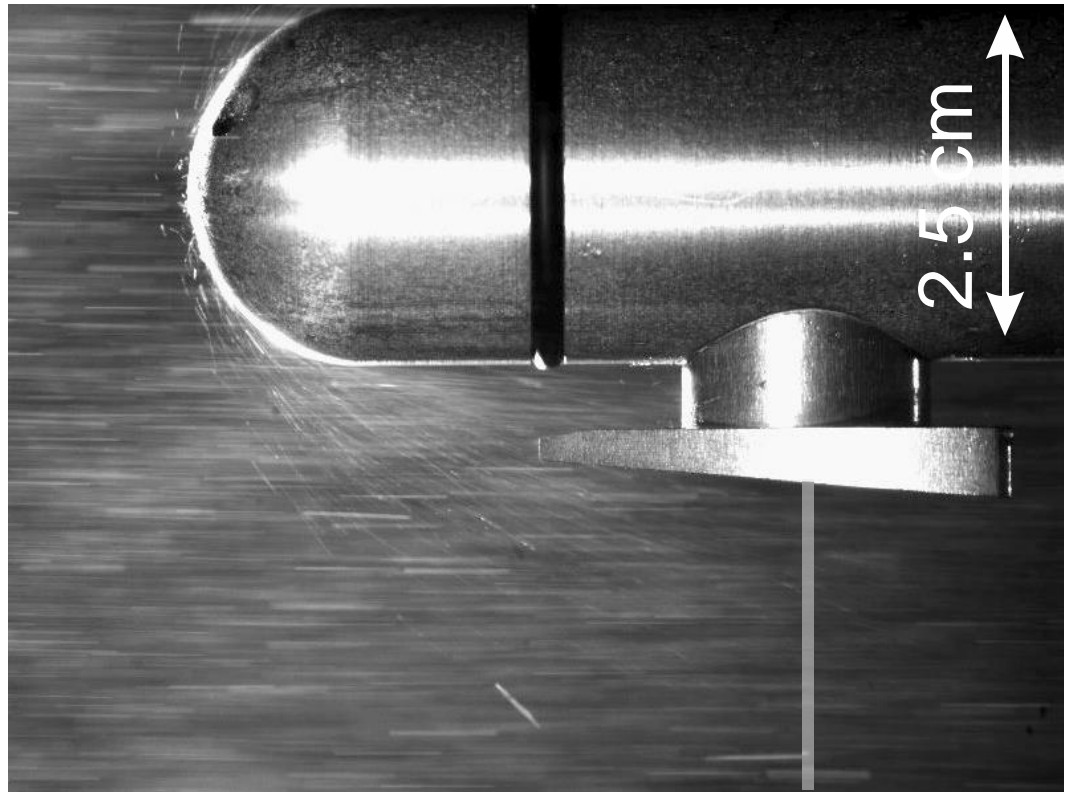

**Figure 15**. A snapshot from a high-speed video showing a flow of shattered ice fragments rebounding from the
hemispherical tip of the OAP-2DC particle probe. The shattered ice fragments deflected toward the sample area (bright vertical band) are counted by the probe and artificially enhance the measured concentration. The video recording was performed in an ice spray at 80m/s in the Cox and Co. Icing Wind Tunnel Facility (for more details see Korolev et al. 2011, 2013ab).

Korolev et al. (2011, 2013a), Lawson (2011), Korolev and Field (2015), Jackson et al. (2015) showed that the effect of shattering can be mitigated by using both antishattering K-tips (Korolev et al. 2013b) and the interarrival time algorithm (Field et al. 2006). It was also demonstrated that the interarrival time algorithm, when used alone, is not capable of identifying all shattering artifacts. Korolev et al. (2013a) showed that measured concentration of ice particles smaller 200μm can be enhanced due to the shattering effect by up to

two orders of magnitude, whereas the concertation of ice particles larger than 400μm remains mainly unaffected.

     Another source of artifacts in measurements of high concentration of ice particles by optical array probes (OAP), is related to fragmentation of particle images when particles pass through the sample volume close to the edge of the depth-of-field (DoF) (Korolev 2007, Guélis eta al., 2019). A few one-two pixel images resulted

from fragmentation of large out-of-focus images have an enhanced artificial contribution into particle concentration due to their very small sample volumes. This is a purely optical phenomenon, and it is relevant only to imaging particle probes. Currently, the problem of fragmented images is recognized by many research groups. One of possible solutions of this problem is the exclusion of the first two or three size bins compromised by the ambiguity of the DoF definition and contamination by image fragments. Due to the extent

that particle counts from the first two or three size bins (smaller than 30 - 80μm depending on the OAP type) may significantly contribute to the ice concentration, a limitation is imposed on the measurements of total concertation of ice particles in SIP cloud regions.

     These findings brought up a question whether early airborne studies of SIP were contaminated by shattering artifacts, which resulted in an artificial enhancement of the measured concentration of small ice. However,

numerous recent in-situ measurements, which employed the antishattering techniques and updated processing algorithms, are in general consistent with the early SIP observations, and they also showed that in many clouds, ice particle concentrations are still much higher than the INP concentration (e.g. Crosier et al. 2011, 2014; Crawford et al. 2012; Heymsfield and Willis, 2014; Stith et al. 2014; Lawson et al, 2015; 2017; Lloyd et al. 2015; Lasher-Trapp et al. 2016; Keppas et al. 2017; Ladino et al. 2017; Korolev et al. 2020; and others).


## 9.   Concluding remarks

### 9.1  General comments

     Figure 16 shows a summary diagram with conceptual models of six SIP mechanisms discussed above.

     The analysis provided in this work shows that the experimental studies SIP are distributed quite unevenly

between different mechanisms. Most of the SIP experimental works are associated with examining the mechanism of droplet fragmentation during freezing (33 publications[2]). A large number of laboratory works

---

[2] A publication is considered related to a specific SIP mechanism, if it includes *experimental* results related to this specific mechanism. Theoretical and in-situ observational works were not counted. Note, that some publications were not cited in this work.

are dedicated to studying the rime splintering (HM-process) (22 publications). The other four mechanisms received far too little attention from the lab research community: ice-ice collisional fragmentation (2 publications), thermal shock fragmentation (3 publications), sublimating ice fragmentation (9 publications); INP nucleation in transient supersaturation (5 publications). Even though none of the above mechanisms have a complete quantitative theoretical description, there is a reasonably good understanding of what physical processes are involved in these mechanisms with the exception of the HM-process. The situation regarding the HM process is contradictory: on one hand, the parameterization of the HM-mechanism is widely used in cloud simulations and weather prediction models, on the other hand, there is no clear understanding of the physical processes underlying this mechanism. At the same time, none of the other five mechanisms is employed on a systematic basis in weather prediction models.

The most striking outcome of this review is the diverse range of results obtained by different research groups for each of the SIP mechanisms. This is one of the major issues hindering the development of physically based parameterizations for numerical simulations. On the other hang the diversity of the results of laboratory studies challenges the existing SIP parameterizations employed in numerical simulations of clouds and weather prediction models.

Droplet fragmentation during freezing

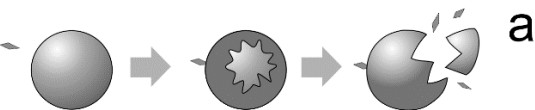

Ice fragmentation during thermal shock

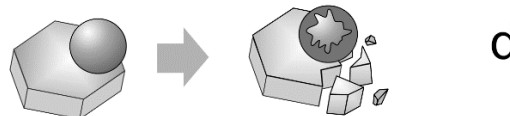

Splintering during riming
(Hallett-Mossop process)

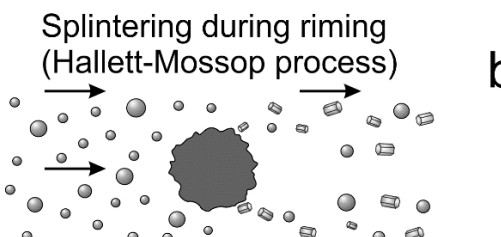

Fragmentation during sublimation

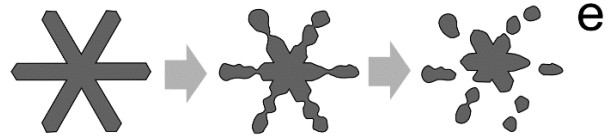

Fragmentation during ice-ice collision

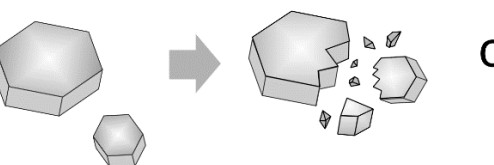

Activation of INP in transient supersaturation

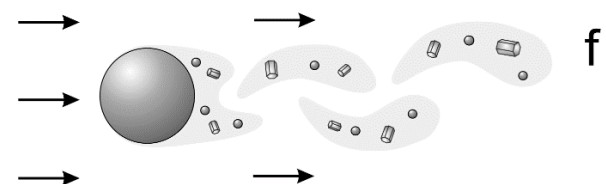

**Figure 16.** A conceptual diagram summarizing six SIP mechanisms (a) fragmentation droplets during freezing; (b) rime splintering (Hallett-Mossop process); (c) fragmentation of ice particles during ice-to-ice collision; (d) fragmentation of ice particles during thermal shock caused by a freezing drop attached to their surfaces; (e) fragmentation of ice particles during their sublimation; (f) activation of supersaturation sensitive INP in the transient supersaturation formed around freezing drops or wet graupel/hailstones.

     **9.2    Feasibility of SIP mechanisms**

One of the important questions related to ice multiplication is whether all six mechanisms can occur in natural clouds.

The review of the lab experiments suggests that the mechanism of droplet fragmentation during freezing may be active across a wide range of temperatures. There is an increasing amount of evidence indicating
universality of this mechanism, which may occur in both convective and stratiform clouds.

The rime splintering (HM) mechanisms require the presence of heavily rimed graupel with high fall velocity. Formation of such graupel is most likely to occur in convective mixed phase cloud regions in a quite narrow temperature range $-8C<T_a<-3C$.

Ice-ice collisional fragmentation requires a large separation of vertical velocities of ice particles to enhance
kinetic energy of their collision. The most likely candidates for this process are lightly and heavily rimed ice particles. The formation of graupel usually occurs in mixed phase convective regions. Whether diffusionally grown ice particles may get fragmented colliding with each other remains unclear.

The theoretical analysis of the thermal shock fragmentation (King and Fletcher, 1976a) suggests that it requires precipitation size drops and temperatures lower than -10°C. Such conditions would be relevant for
mixed phase convective cloud regions where large drops could be transported by a vertical updraft to levels with low temperatures.

Activation of SIP due to the fragmentation of sublimating ice requires spatial proximity of undersaturated and supersaturated cloud regions. In this case, secondary ice particles formed in the undersaturated cloud regions can be rapidly transported into the supersaturated regions prior their sublimation. Such conditions may
occur in cloud regions affected by entrainment and mixing with out-of-cloud dry air.

INPs activation in transient supersaturation requires precipitation size drops and high supercooling. As indicated above, such conditions are typical for convective cloud regions.

Out of six SIP mechanisms, the droplet fragmentation during freezing and INP activation in transient supersaturation mechanisms appear to be primary candidates for initial production of secondary ice at the early
stage of ice formation in convective clouds. The rest of the mechanisms require preexisting aged ice, and they may contribute to the ice concentration at later stages of cloud development.

It is worth mentioning that the possibility of additional SIP mechanisms beyond the purview of this paper (Fig.16) remains unexplored. In this regard, studying the existence of other SIP mechanisms not described in this study is still on the agenda of SIP investigation (e.g. Knight 2012).
When discussing the feasibility of SIP mechanisms, it is important to keep in mind well documented observations of supercooled persistent mixed-phase clouds with temporally stable low ice concentration ($<0.5$-$5L^{-1}$) (e.g. Korolev et al. 2017; McFarquhar et al. 2011; Shupe et al., 2006). In these clouds, seemingly satisfying some conditions required for SIP, no explosive enhancement of ice concentration was observed.

Similar mixed-phase cloud environments with no SIP were also reproduced in laboratory experiments (Desai et al. 2006). These in-situ and laboratory observations accentuate the importance of identifying the necessary and sufficient conditions required for the initiation of each of the SIP mechanisms.

Another unexplored possibility is related to enhancing the activation properties of typically ineffective primary INPs due to changing the local properties of the cloud environment. In section 7, such activation of ineffective primary INPs occurred in the cloud environment modified by freezing drops (or wet hail and rimed ice particles) due to local increase of supersaturation.

### 9.3 The way forward

The large discrepancies within the experimental results obtained by different research groups necessitates the development of laboratory setups that account for a variety of possible parameters that may be implicated in different SIP processes.

Because of the complexity involved in researching SIP, obtaining consistent results from independent research groups is an important task for SIP studies. This would require consolidating efforts across the cloud physics community at the international level (Shaw et al., 2020). Laboratory investigations should go hand in hand with the development of theoretical descriptions of the SIP processes on a microscale level and in-situ observations. This will create a foundation for physically based parameterizations for weather and climate models, which is the ultimate goal of all these efforts.

*Authors' contribution:* AK and TL prepared the manuscript.

*Competing interests.* The authors declare that they have no conflict of interest.

*Acknowledgments*: This work was supported by Environment and Climate Change Canada (ECCC), Transport Canada (TC), the USA Federal Aviation Administration (FAA) funds and by the Helmholtz Association under the Atmosphere and Climate Program (ATMO). Special thanks to the ECCC librarians Danny Chan and Derek Funston for outstanding support in sourcing literature and provision of publications referenced in this study. The authors are grateful to Jim Dye (NCAR) and Paul Lawson (SPEC) for diligent and inspired discussion. The authors wish to express their gratitude to the open referees Andy Heymsfield (NCAR) and Raymond Shaw (MTI) for their thorough reviews and valuable comments.

**Appendix A**: List of symbols and abbreviations

| Symbol | Description |
|---|---|
| $D$ | droplet diameter |
| $D_{eff}$ | droplet effective diameter |
| $D_v$ | water vapor diffusion coefficient |
| $c_i$ | specific heat of ice |
| $c_w$ | specific heat of liquid water |
| $f$ | ventilation coefficient |
| $G$ | non-rational growth velocity of ice in liquid water |
| $G_a$ | velocity of ice growth in liquid water along the $a$-axis |
| $G_c$ | velocity of ice growth in liquid water along the $c$-axis |
| $K$ | thermal conductivity of surrounding medium |
| $K_a$ | thermal conductivity of the air |
| $L_m$ | latent heat of freezing, |
| $L_s$ | latent heat of ice sublimation |
| $m$ | mass |
| $P$ | air pressure |
| $r_c$ | average critical droplet radius separating droplet freezing as single-crystal or polycrystal |
| $RH_i$ | relative humidity over ice |
| $RH_w$ | relative humidity over water |
| $t_1$ | time scale of the recalescence (fast) stage during droplet freezing |
| $t_2$ | time scale of the freezing (slow) stage during droplet freezing |
| $T_a$ | air temperature |
| $T_m$ | ice melting temperature |
| $T_n$ | ice nucleating temperature |
| $T_s$ | droplet or rimer surface temperature |
| $u_z$ | free fall velocity |
| $\Delta m$ | liquid water mass frozen during recalescence stage |
| $\Delta T$ | $T_m - T_a$ supercooling |
| $\mu$ | $\Delta m/m$, liquid fraction frozen during recalescence stage |
| $\rho_a$ | air density |
| $\rho_r$ | rimer density |
| $\rho_v$ | water vapor density |
| $\rho_w$ | liquid water density |
| CIP | Cloud Imaging Probe |
| DNS | Direct Numerical Simulation |
| DoF | Depth-of -Field |
| EDB | Electro-Dynamic Balance |
| INP | Ice Nucleating Particles |
| LWC | Liquid Water Content |
| OAP | Optical Array Probe |
| SIP | Secondary Ice Production |

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
