# Peer review of "Review of experimental studies of secondary ice production"

_Atmospheric Chemistry and Physics, 2020_

## Referee Comment (RC1) · Andrew Heymsfield (Referee) · 10 Jul 2020

This is an excellent review article of a process that is very important for precipitation development, Secondary Ice Production (SIP). Based on observations I've made and those reported on by others, the process is particularly important when 1) cloud top temperatures are relatively warm, 2) there are relative few but some ice nuclei active at the cloud temperatures, 3) the cloud droplet sizes are relatively large, and the updraft velocities, although present, and not too strong, one to a few meters per second. SIP is therefore likely to be most active and important over relatively warm oceanic areas. Although several SIP mechanisms have been proposed, it is unclear when specific ones are active and under what conditions they occur.

[Figure]

This article discusses the plusses and minuses of the following SIP that have been proposed: (1) shattering during droplet freezing; (2) the rime splintering (Hallett-Mossop) process; (3) fragmentation due to ice-ice collision; (4) ice particle fragmentation due to thermal shock; (5) fragmentation of sublimating ice; and (6) activation of ice nucleating particles in transient supersaturation around freezing drops. The article focuses on laboratory studies, although some field observations are presented. Obviously, laboratory studies benefit from the ability to repeat experiments and narrow down possible processes by modifying the experiments appropriately.

I have relatively few comments because the article is very well written and extremely thorough.

Line 34: Schaefer

Line 59: "shattering" to "fragmentation

Section 2.2, Eqs. (1) and (2), Freezing Fraction. Shouldn't ventilation enter into this discussion? It is factored into Eq. (3).

Section 2.7, Fragmentation during freezing. A table summarizing your discussion of shattering during drop/droplet freezing would be very helpful.

Line 445: During the freezing process, the surface of the droplet is sublimating, perhaps affecting the fragmentation process.

Line 649: undersaturated>subsaturated.

Line 732 "bigger" to "larger".

Line 783: the existence of shattering.

Line 824: concentration.

Line 825: You could mention the airborne studies of Mossop and Bigg and the use of balloon borne replicators, etc that can shed light on the problem. Many other airborne

studies (Heymsfield and Willis, Lawson et al., Lasher-Trapp etc are directed towards the SIP problem. At the beginning of this section, re-emphasize that this review article is mostly directed towards laboratory and theoretical studies.

Andy Heymsfield, NCAR

---

## Referee Comment (RC2) · Raymond Shaw (Referee) · 25 Jul 2020

**Raymond Shaw (Referee)**

rashaw@mtu.edu

Received and published: 25 July 2020

This review article fills an important void in the atmospheric chemistry and physics literature by aggregating and synthesizing the broad range of literature relevant to secondary ice production in clouds. The article is thorough in its review of the literature and is helpful in going beyond merely reporting prior results, but placing them within the context of the full body of work and the current understanding. I had two personal impressions while reading the review. First, we as a community have strayed too far from our roots, and while there is some excellent laboratory work still taking place, it is disproportionately small compared to the vast efforts currently focused on field and computational work. I agree with the authors' perhaps provocative statement that laboratory work cannot be replaced by field work if we hope to achieve physically-based

understanding and parameterization of SIP processes. Rather, these efforts need to take place hand in hand. Second, there is clear value in bringing all of the relevant experimental results together for a cohesive review, with the result being much more impactful than simply the sum of individual studies. Meaningful theoretical progress rests on the observations from these collective experiments. To put it another way, taking electromagnetism as an example, there would have been no Maxwell without a Faraday. I hope these impressions come through clearly to other readers of this review. I would go so far as to say that the authors should take the liberty of editorializing somewhat more along these lines in the Concluding Remarks section; that is their prerogative, though, and I only share it as my opinion. In any case, I consider this to be a well written and important contribution that will help in providing a deeper understanding of what we know about SIP, and motivation for further work on this topic. Figure 16 is a distillation of the key findings and illustrates the mechanisms in a clear, graphical way that should help the main results be accessible to a broad range of readers, including those from other communities who may have overlapping interests with the subject (e.g., materials science, turbulence, etc.). Variations of this figure will likely appear in future cloud physics textbooks.

The following suggestions should be considered in revising the review. I have listed them roughly in order of priority.

1. In the "Way Forward" section it would be very helpful to summarize some of the key points that came up throughout the paper, regarding what aspects need to be carefully considered in future laboratory experiments. I came up with the following list, but may have missed some points. Laboratory experiments on SIP mechanisms likely need to consider the following variables, in order to ensure that the results are of atmospheric relevance:

- Particle fall speed and its influence on enhancement of diffusive fluxes, mixing in turbulent wake (ventilation effects).
- Relative velocity and impact parameter for mechanisms involving particle-particle interactions.

- Ambient atmospheric pressure, temperature and gas properties.

- Thermal equilibration of particles with the surrounding atmosphere (or realistic values of thermal lag for typical atmospheric temperature profiles and turbulence properties).

- Presence of dissolved gases and other impurities; in particular, unrealistically high concentrations of gases such as CO2 should be avoided.

2. One possibility not considered in defining primary vs secondary ice production: Could there be "primary" mechanisms that do not involve INP, or that strongly enhance properties of otherwise ineffective INP? Might these be relevant since they could lead to apparent discrepancy between expected and observed number of ice crystals, even without the operation of secondary ice processes? I have in mind pressure perturbations and electrical effects, as examples (citing work with which I am familiar... I am sure there is much more... consider Yang et al. 2015, Applied Physics Letters "Ice nucleation at the contact line triggered by transient electrowetting fields" and Yang et al. 2018, Phys Rev E "Nonthermal ice nucleation observed at distorted contact lines of supercooled water drops").

3. It also would seem relevant to discuss laboratory or cloud chamber experiments in which no secondary ice production was observed. Knowing the conditions under which SIP is not required is valuable for determining what part of parameter space should be searched. Again, drawing from familiar work, and acknowledging that there must be more, I have in mind the cloud chamber study of Desai et al. 2019, GRL "Aerosol-Mediated Glaciation of Mixed-Phase Clouds: Steady-State Laboratory Measurements", in which agreement was observed between injected INP and observed ice crystal concentration, to within experimental uncertainties. Interestingly, although it was not emphasized in the paper, multiple images of "pac-man" shaped ice crystals were observed in that study (see their Figure 1), and one can speculate that ACPD
they are fragmented frozen cloud droplets. And yet they do not seem to have contributed significantly to ice budgets under the existing experimental conditions (limited to relatively small particle sizes with only 1 meter of vertical distance for fallout and thereby limited particle lifetimes).

4. My first impression was that there is a lack of balance between the ice shattering mechanism covered in section 2 compared to the other sections. Upon reflection, though, I realize that it is a result of more literature being available in that area. A word on this at the beginning of section 2 would help orient the reader, allowing to understand that the fundamentals of ice growth in supercooled liquid, etc., have been thoroughly studied and are of relevance to the drop shattering problem covered later in the section.

5. The discussion on page 4 (especially near lines 391-392) raises the question of gas equilibration time. For natural cloud droplets grown by condensation the gas content may be substantially different than for droplets generated in a laboratory from atomizing a bulk liquid that is presumably in equilibrium with ambient gases. Could this be a relevant factor to consider?

6. Clarify on lines 58-61 that you are referring to artificial ice shattering that results from sampling/measurement (as opposed to ice shattering from natural processes, which are also considered in this review).

7. Clarify that equation 4 is for the assumption of a spherical droplet in air. Also, for people in other fields who might be accessing this review, provide a reference for the ventilation coefficient and specify that it is a function of terminal speed and therefore of diameter.

8. Clarify on lines 176-178 what is meant by the "spatial scale of the ice crystals".

9. The meaning of Figure 5 and "the diameter of the monocrystalline frozen drops decreases with the increase of supercooling" is not clear to me. How do I interpret a

**ACPD**
data point at a specific radius and supercooling? Does it mean that at lower supercoolings the frozen drops are single crystals and at higher supercoolings the drops are polycrystals? I would have assumed that there is a probability of single versus multiple crystals. Is the data point the probability of 0.5? More explanation is needed here.

10. In a few places there should be more acknowledgement of uncertainty, such as line 275 where it is probably more reasonable to say "Such a high rate of splinter production may be an important factor in the INP economy"... since it surely depends on many other factors as well. On the other hand, I see at least one place where the view of the field may be overly pessimistic: line 432 "remains poorly understood or unknown" would seem more reasonable to be "remains only partially understood."

11. In the caption of Figure 10 you refer multiple times to INP, but in this case you are referring to an ice particle colliding with a supercooled droplet. Strictly speaking, yes, the ice particle could be considered an INP, but to me it seems misleading. If we consider INP as usually defined, then this refers more to heterogeneous (primary) ice nucleation. Indeed the question of surface versus volume crystallization is intriguing, but it is more closely related to primary ice formation.

12. In the paragraph discussing the paper of Baker (1991), near line 750, it should be made clearer that Baker considered static drops during the transient freezing process, whereas others such as Prabhakaran et al. account for continuous production of supersaturation in the wake of a falling particle that is riming (wet growth) or melting. In the next paragraph I would also suggest that it would be helpful to have a better understanding of how INP behave at very high liquid-water supersaturations, since this is a regime not typically achieved with current instruments (Fukuta had a wedge method that produced very high supersaturations and indeed observed higher INP efficiency in that regime).

13. My initial reaction was that Section 8 does not really fit with the main theme of the review. It is relevant in the sense that spurious ice crystals may have contributed to

**ACPD**
the apparent conflict between measured INP and measured ice crystal concentrations. But then it begs the question why other field measurements are not reviewed as well. One aspect that could be emphasized to strengthen the connection to the laboratory focus of the paper is that the high speed videos in the Koroleve et al. papers were obtained in a wind tunnel setting (at least that is my recollection). Perhaps this is a good place to emphasize that lab "experiments" have contributed not only to understanding of fundamental mechanisms, but also to the evaluation of measurement techniques applied in the field. Those videos captured in a controlled lab environment settled the question of shattered ice crystals in the minds of many in the community. I would also point out that this section misses an important reference to Jackson et al. 2014, JAOTech, who made a full assessment of measurement-induced ice shattering based on intercomparison of multiple instruments.

14. In Section 9.1 it could be useful to elaborate more on "the most striking outcome of this review", that there is such a wide range of results for each SIP mechanism. In conversation with colleagues I have encountered a sense of exasperation that lab experiments sometimes show bewildering complexity. I even remember a story from an individual involved with experiments in the Hobbs lab in the late '60s that suggested that the CO2 contamination they identified as a cause of drop shattering is one factor that motivated Prof. Hobbs to shift his group's emphasis to field work. The result of the "striking outcome", however, should be that we carry out more, not less experimental work, in order to clarify the various unrecognized factors and ultimately to gain a full understanding of the relevant processes.

A thorough check for grammatical and typographical errors should be made. Overall the writing is excellent, but there are multiple places where small errors appear. I summarize the ones I found, although I probably did not catch all while reading:

Title: It sounds more natural to my ear to say "experimental studies of secondary ice production" rather than "on secondary ice production". But I'm not a grammar expert, so I would not go so far as to say it is incorrect.
Line 34: Schaefer

Caption of Figure 4: \Delta T = 14.5 C (should not be negative as currently shown).

Line 205 and several other places: A temperature is high or low (not warm or cold, which is only for an object).

Line 208: the Visagie experiments.

Lines 335-336: experiments that had a droplet suspended (no "is" needed).

Caption of Figure 8: I do not find Lauber et al. 2015 in the references. Should it be 2018?

Line 425: number of parameters.

Line 489: in a cloud chamber.

Line 511: studies of Hallett and Mossop.

Line 526: rapid growth of the ice shell.

Line 581: Phillips (also this sentence might be clearer is written "The studies of Hobbs and Farber, Vardiman, and Phillips et al. were based on the consideration...").

Line 633: thickness of.

Caption of Figure 14: Agl and Snomax were used as ice nucleating particles.

Lines 722-723: aerosols introduced into the ambient air.

Line 737: ice sphere in humid air.

Line 738: The word "study" appears twice... should be rephrased.

Line 765: Not sure if "feedbacking" is a word.

Line 783: the existence of shattering.

**ACPD**
Line 843: a complete quantitative theoretical description.

Line 848: systematic basis in weather prediction models.

---

## Author Comment (AC1) · 26 Aug 2020

**Replies to the reviewer's comments (Andrew Heymsfield) on "Review of experimental studies on secondary ice production" by A. Korolev and T. Leisner**

**From Authors**: The authors appreciate the reviewer's time spent to read the paper and provide a diligent review. We found the comments very helpful in improving the manuscript. Below are point-by-point replies to the comments.

This is an excellent review article of a process that is very important for precipitation development, Secondary Ice Production (SIP). Based on observations I've made and those reported on by others, the process is particularly important when 1) cloud top temperatures are relatively warm, 2) there are relative few but some ice nuclei active at the cloud temperatures, 3) the cloud droplet sizes are relatively large, and the updraft velocities, although present, and not too strong, one to a few meters per second. SIP is therefore likely to be most active and important over relatively warm oceanic areas. Although several SIP mechanisms have been proposed, it is unclear when specific ones are active and under what conditions they occur.

This article discusses the plusses and minuses of the following SIP that have been proposed: (1) shattering during droplet freezing; (2) the rime splintering (Hallett-Mossop) process; (3) fragmentation due to ice-ice collision; (4) ice particle fragmentation due to thermal shock; (5) fragmentation of sublimating ice; and (6) activation of ice nucleating particles in transient supersaturation around freezing drops. The article focuses on laboratory studies, although some field observations are presented. Obviously, laboratory studies benefit from the ability to repeat experiments and narrow down possible processes by modifying the experiments appropriately.

I have relatively few comments because the article is very well written and extremely thorough.

Line 34: Schaefer
**Reply**: Corrected.

Line 59: "shattering" to "fragmentation
**Reply**: Corrected.

Section 2.2, Eqs. (1) and (2), Freezing Fraction. Shouldn't ventilation enter into this discussion? It is factored into Eq. (3).
**Reply**: The ventilation factor in Eq.1 is included in the term $\Delta Q$ which describes the heat loss due to thermal exchange with the environment. Due to a very short duration of the recalescence stage ($10^{-5}$s$<t_1<10^{-1}$s depending on $D$ and $\Delta T$ ) the thermal exchange between the surrounding environment and droplet is much smaller compared to the energy of the latent heat released during freezing. Accurate assessment of $\Delta Q$ suggests that it is much smaller other terms in Eq.1 and it is usually neglected. Eq.3 employs $u(T)$ obtained from experimental measurements. As before the due to a very short time of the recalescence stage the effect of ventilation in calculation of $t_1$ is neglected.

Section 2.7, Fragmentation during freezing. A table summarizing your discussion of shattering during drop/droplet freezing would be very helpful.
**Reply**: Table 1 summarizing shattering during drop/droplet freezing was added in the text following the reviewer's comment. In addition, we also added Table 2 summarizing laboratory results of the studies of the HM-process.

**Table 1**. Summary of experimental studies of droplet fragmentation during freezing by different research groups. The table covers only works that quantified the parameters included in the table.

| Reference | Diameter (mm) | Temperature (°C) | Droplet suspension | Method of nucleation | Maximum SIP frequency (%) | max number fragments per drop | Temperature of maximum SIP rate |
|---|---|---|---|---|---|---|---|
| Mason and Maybank, 1960 | 30-1000 | -2 to -25 [1] | stagnant (fiber) | various [2] | 47 | 200 | -10C |
| Adkins, 1960 | 4-13 | n/a [3] | free fall | natural [4] | 0 | 0 | n/a |
| Hobbs and Alkezweeny, 1968 | 20-150 | -8 to -32 | free fall | various [5] | >5 | n/a | no temperature dependence |
| Brownscombe and Thorndike, 1968 | 50-90 | -5, -10, -15 | free fall | tiny ice crystals | 14 | 12 | -15°C |
| Dye and Hobbs, 1968 | 1000 | -3 to -15 | stagnant (fiber) | tiny ice crystals | 0 | 1 | no temperature dependence |
| Johnson and Hallett, 1968 | 1000 | -5 to -20 | stagnant (fiber) +ventilation | tiny ice crystals | >1 | n/a | no temperature dependence |
| Takahashi and Yamashita, 1969 | 600-800 | -18 to -25 | free fall | immersion [6] | 11 | n/a | -15°C |
| Takahashi and Yamashita, 1970 | 75-350 | 0 to -30 | free fall | tiny ice crystals | 37 | n/a | -15°C |
| Takahashi, 1975 | 45-765 | -4 to -24 | free fall | tiny ice crystals | 35 | n/a | -16°C |
| Pruppacher and Schlamp, 1975 | 410 | -7 to -23 | airflow | contact [7] | 15 | >3 | -11°C to -15°C |
| Bader et al., 1974 | 30, 42, 84 [8] | -10 to -30 | free fall | immersion [9] | ? | 10 | n/a |
| Kolomeychuk et al., 1975 | 1600 | -12 to -25 | airflow [10] | natural [4] | 35 | 142 | -15°C to -18°C |
| Lauber et al., 2018 | 300 -320 | -5 to -30 | stagnant (EDB) | tiny ice crystals | 35 | 12 | -7C to -13C |
| Keinert et al., 2020 | 300- 320 | -1 to-30 | stagnant (EDB), airflow | tiny ice crystals | 1 | 3 | -10C to -15C |

1. ice nucleation temperature 0°C to -15°C
2. natural nucleation, silver iodide, contact tiny small ice crystals
3. not available
4. no special efforts were made to nucleate droplets
5. natural or immersed silver iodide
6. kaolinite or silver iodide
7. kaolinite or montmorillonite
8. mean volume diameter
9. silver iodide
10. flow of humidified nitrogen

Line 445: During the freezing process, the surface of the droplet is sublimating, perhaps affecting the fragmentation process.

**Reply**: The referee is right, roughly 1 % of the droplet mass evaporate per 1K of initial supercooling. This is a small amount and most of this evaporation occurs during the early stages of the second freezing process (Keinert et al. 2020), while the SIP processes occur during the later stages of the second freezing process. It seems, that it is unlikely that evaporation will affect SIP much, and therefore, we do not discuss it in the manuscript.

Line 649: undersaturated>subsaturated.
**Reply**: Corrected.

Line 732 "bigger" to "larger".
**Reply**: Corrected.

Line 783: the existence of shattering.
**Reply**: Corrected.

Line 824: concentration.
**Reply**: Corrected.

Line 825: You could mention the airborne studies of Mossop and Bigg and the use of balloon borne replicators, etc that can shed light on the problem. Many other airborne studies (Heymsfield and Willis, Lawson et al., Lasher-Trapp etc are directed towards the SIP problem. At the beginning of this section, re-emphasize that this review article is mostly directed towards laboratory and theoretical studies.
**Reply**: Several early Mossop's airborne studies (1970, 1972, 1985) were reverenced in the introduction. The references on the studies of Heymsfield and Willis, Lawson et al., Lasher-Trapp et al. were provided in the concluding section to address the reviewer's comment.
The Bigg (1996) studied IFNs in Arctic clouds. There is a brief mentioning of SIP among other possible reasons to explain one of the observations of discrepancy between ice particle and IFN concentrations. This does not provide strong evidences of SIP compared to the early studies, such as that by Koenig, Hobbs, Mossop, and others mentioned in the introduction. The authors consider that referencing of the Bigg (1996) study would be destructive as not directly relevant to this study.
At the beginning of the section 9 the following text was added to address the reviewer's comment regarding laboratory studies:
"In this section we discuss results of experimental studies of artificial fragmentation of ice particles during in-situ sampling."

---

## Author Comment (AC2) · 26 Aug 2020

**Replies to the reviewer's comments (Raymond Shaw) on "Review of experimental studies on secondary ice production" by A. Korolev and T. Leisner**

**From Authors**: The authors appreciate the reviewer's time spent to read the paper and provide a diligent review. We found the comments very helpful in improving the manuscript. Below are point-by-point replies to the comments.

This review article fills an important void in the atmospheric chemistry and physics literature by aggregating and synthesizing the broad range of literature relevant to secondary ice production in clouds. The article is thorough in its review of the literature and is helpful in going beyond merely reporting prior results, but placing them within the context of the full body of work and the current understanding. I had two personal impressions while reading the review. First, we as a community have strayed too far from our roots, and while there is some excellent laboratory work still taking place, it is disproportionately small compared to the vast efforts currently focused on field and computational work. I agree with the authors' perhaps provocative statement that laboratory work cannot be replaced by field work if we hope to achieve physically-based understanding and parameterization of SIP processes. Rather, these efforts need to take place hand in hand. Second, there is clear value in bringing all of the relevant experimental results together for a cohesive review, with the result being much more impactful than simply the sum of individual studies. Meaningful theoretical progress rests on the observations from these collective experiments. To put it another way, taking electromagnetism as an example, there would have been no Maxwell without a Faraday. I hope these impressions come through clearly to other readers of this review. I would go so far as to say that the authors should take the liberty of editorializing somewhat more along these lines in the Concluding Remarks section; that is their prerogative, though, and I only share it as my opinion. In any case, I consider this to be a well written and important contribution that will help in providing a deeper understanding of what we know about SIP, and motivation for further work on this topic. Figure 16 is a distillation of the key findings and illustrates the mechanisms in a clear, graphical way that should help the main results be accessible to a broad range of readers, including those from other communities who may have overlapping interests with the subject (e.g., materials science, turbulence, etc.). Variations of this figure will likely appear in future cloud physics textbooks.

The following suggestions should be considered in revising the review. I have listed them roughly in order of priority.

1. In the "Way Forward" section it would be very helpful to summarize some of the key points that came up throughout the paper, regarding what aspects need to be carefully considered in future laboratory experiments. I came up with the following list, but may have missed some points. Laboratory experiments on SIP mechanisms likely need to consider the following variables, in order to ensure that the results are of atmospheric relevance:
- Particle fall speed and its influence on enhancement of diffusive fluxes, mixing in turbulent wake (ventilation effects).
- Relative velocity and impact parameter for mechanisms involving particle-particle interactions.
- Ambient atmospheric pressure, temperature and gas properties.
- Thermal equilibration of particles with the surrounding atmosphere (or realistic values of thermal lag for typical atmospheric temperature profiles and turbulence properties).
- Presence of dissolved gases and other impurities; in particular, unrealistically high concentrations of gases such as $CO_2$ should be avoided.

**Reply**: Because of pronounced differences between the parameters controlling production of secondary ice for each of the six mechanisms, the authors consider that summarizing all parameters in one list or table would be not an easy task or even confusing. For example, concentration of $CO_2$ dissolved in water plays an important role for droplets fragmentation during freezing. However, it hardly has any effect on fragmentation during ice-ice collision or ice sublimation. In order to address the reviewer's comment, it was decided to summarize the list of parameters, which may affect secondary ice production, at the end of the sections related to each specific SIP mechanism.

2. One possibility not considered in defining primary vs secondary ice production: Could there be "primary" mechanisms that do not involve INP, or that strongly enhance properties of otherwise ineffective INP? Might these be relevant since they could lead to apparent discrepancy between expected and observed number of ice crystals, even without the operation of secondary ice processes? I have in mind pressure perturbations and electrical effects, as examples (citing work with which I am familiar... I am sure there is much more to consider Yang et al. 2015, Applied Physics Letters "Ice nucleation at the contact line triggered by transient electrowetting fields" and Yang et al. 2018, Phys Rev E "Nonthermal ice nucleation observed at distorted contact lines of supercooled water drops").

**Reply**: According to our definition, electrowetting induced freezing (Yang et al. 2015, 2018) would be classified as primary ice production as it does not require pre-existing ice particles. We rather want to stick to our narrow definition of secondary ice production here. There are border line cases however and the following paragraph was added in section 9.2 to address the reviewer's comment:

"Another unexplored possibility is related to enhancing the activation properties of typically ineffective primary INPs due to changing the local properties of the cloud environment. In section 7, such activation of ineffective primary INPs occurred in the cloud environment modified by freezing drops (or hail and rimed ice particles) due to local increase of supersaturation."

3. It also would seem relevant to discuss laboratory or cloud chamber experiments in which no secondary ice production was observed. Knowing the conditions under which SIP is not required is valuable for determining what part of parameter space should be searched. Again, drawing from familiar work, and acknowledging that there must be more, I have in mind the cloud chamber study of Desai et al. 2019, GRL "Aerosol Mediated Glaciation of Mixed Phase Clouds: Steady State Laboratory Measurements", in which agreement was observed between injected INP and observed ice crystal concentration, to within experimental uncertainties. Interestingly, although it was not emphasized in the paper, multiple images of "pac-man" shaped ice crystals were observed in that study (see their Figure 1), and one can speculate that they are fragmented frozen cloud droplets. And yet they do not seem to have contributed significantly to ice budgets under the existing experimental conditions (limited to relatively small particle sizes with only 1 meter of vertical distance for fallout and thereby limited particle lifetimes).

**Reply**: The following paragraph was added in section 9.2 to address the reviewer's comment:

"When discussing the feasibility of SIP mechanisms, it is important to keep in mind well documented observations of supercooled persistent mixed-phase clouds with temporally stable low ice concentration ($<1$-$5L^{-1}$) (e.g. Korolev et al. 2017; McFarquhar et al. 2011; Shupe et al., 2006). In these clouds, seemingly satisfying some conditions required for SIP, no explosive enhancement of ice concentration was observed. Similar mixed-phase cloud environments with no SIP were also reproduced in laboratory experiments (Desai et al. 2006). These in-situ and laboratory observations accentuate the importance of identifying the necessary and sufficient conditions required for the initiation of each of the SIP mechanisms."

4. My first impression was that there is a lack of balance between the ice shattering mechanism covered in section 2 compared to the other sections. Upon reflection, though, I realize that it is a result of more literature being available in that area. A word on this at the beginning of section 2 would help orient the reader, allowing to understand that the fundamentals of ice growth in supercooled liquid, etc., have been thoroughly studied and are of relevance to the drop shattering problem covered later in the section.

**Reply**: The following text was added at the end of Introduction (section 1) to address the reviewer's comment:

"The authors would like to acknowledge the length disproportions between the aforementioned sections. Section 2 has the biggest volume, which is a reflection of the large amount on knowledge accumulated on different aspects of water freezing and is directly linked to the secondary ice formation during droplet freezing. The rest of the sections have smaller sizes due to fewer laboratory experiments related to them. These disproportions will be discussed in more detail in section 9.1."

5. The discussion on page 4 (especially near lines 391-392) raises the question of gas equilibration time. For natural cloud droplets grown by condensation the gas content may be substantially different than for droplets generated in a laboratory from atomizing a bulk liquid that is presumably in equilibrium with ambient gases. Could this be a relevant factor to consider?

**Reply**: Neglecting chemical reactions inside the droplet, the timescale for gas equilibration for a cloud droplet of 100μm dia. is on the order of magnitude of 10s. This is assumed to be faster than the time required for the droplet to grow to that size, in particular as the trace gases are taken up concurrently with the water vapor during droplet growth and part of the growth process might be the aggregation of smaller saturated droplets. We therefore assume that cloud droplets – like laboratory droplets – are well equilibrated. We do mention dissolved gases as one of the possible "hiden parameters" in lab experiments in the summary of section 2.

6. Clarify on lines 58-61 that you are referring to artificial ice shattering that results from sampling/measurement (as opposed to ice shattering from natural processes, which are also considered in this review).

**Reply**:  The mentioned statement was modified to address the reviewer's comment:
"For the sake of thoroughness, experimental studies of the effects of artificial ice particle fragmentation during in-situ sampling were included in this review as well."

7. Clarify that equation 4 is for the assumption of a spherical droplet in air. Also, for people in other fields who might be accessing this review, provide a reference for the ventilation coefficient and specify that it is a function of terminal speed and therefore of diameter.

**Reply**: We have modified the respective paragrapho read:

"According to Pruppacher and Klett (1998), the duration of the second stage of the droplet freezing inward can be estimated for a spherically symmetric droplet falling in air as:

$$t_2 = \frac{\rho_w L_m D^2 (1 - \frac{\Delta T c_w}{L_m})}{12 f \Delta T \left( k_a + L_s D_v \left( \frac{d \rho_v}{d T} \right)_{sat,i} \right)} \tag{4}$$

where $\rho_w$ liquid water density; $f$ ventilation coefficient; $D_v$ is the water vapor diffusion coefficient; $k_a$ is the thermal conductivity of the air; $L_s$ latent heat of ice sublimation; $\left( \frac{\overline{d \rho_v}}{d T} \right)_{sat,i}$ is the mean slope of the ice saturation vapor density curve over the interval from $T_0$ to $T_m$. The ventilation coefficient f describes the acceleration of droplet freezing from forced (due to the velocity between droplet and gas) and free (due to the temperature difference between droplet and gas) convection as compared to stagnant air (f=1). For drizzle sized droplets falling freely in air, f will typically assume values between 2 and 4."

8. Clarify on lines 176-178 what is meant by the "spatial scale of the ice crystals".
**Reply**: The statement about spatial scale was modified as follows:
 "One of the important findings of studies on water freezing is that the density of the ice mesh increases with the decrease of temperature, whereas the typical size of the ice crystals perpendicular to $a$-axis becomes smaller."

9. The meansing of Figure 5 and "the diameter of the monocrystalline frozen drops decreases with the increase of supercooling" is not clear to me. How do I interpret a data point at a specific radius and supercooling? Does it mean that at lower supercoolings the frozen drops are single crystals and at higher supercoolings the drops are polycrystals? I would have assumed that there is a probability of single versus multiple crystals. Is the data point the probability of 0.5? More explanation is needed here.
**Reply**: In order to clarify the diagram in Fig.5 the following modification of the related text was introduced:
"The average critical radius of a droplet frozen as monocrystalline decreases with the increase of supercooling $\Delta T$ (Fig.5), …"
The caption to Fig.5 was also modified as:
"Dependence of the polycrystallinity of frozen droplets on the average droplet size and freezing temperature of droplets."

10. In a few places there should be more acknowledgement of uncertainty, such as line 275 where it is probably more reasonable to say "Such a high rate of splinter production may be an important factor in the INP economy"… since it surely depends on many other factors as well. On the other hand, I see at least one place where the view of the field may be overly pessimistic: line 432 "remains poorly understood or unknown" would seem more reasonable to be "remains only partially understood."
**Reply**: Both statements were modified as per reviewer's suggestion.
Diligent
11. In the caption of Figure 10 you refer multiple times to INP, but in this case you are referring to an ice particle colliding with a supercooled droplet. Strictly speaking, yes, the ice particle could be considered an INP, but to me it seems misleading. If we consider INP as usually defined, then this refers more to heterogeneous (primary) ice nucleation. Indeed, the question of surface versus volume crystallization is intriguing, but it is more closely related to primary ice formation.
**Reply**: In the caption of Figure 10 the terms "INP" were replaced by "INP or ice crystal" to eliminate potential confusion and address the reviewer's comment.

12. In the paragraph discussing the paper of Baker (1991), near line 750, it should be made clearer that Baker considered static drops during the transient freezing process, whereas others such as Prabhakaran et al. account for continuous production of supersaturation in the wake of a falling particle that is riming (wet growth) or melting. In the next paragraph I would also suggest that it would be helpful to have a better understanding of how INP behave at very high liquid-water supersaturations, since this is a regime not typically achieved with current instruments (Fukuta had a wedge method that produced very high supersaturations and indeed observed higher INP efficiency in that regime).
**Reply**: Baker (1991) makes order of magnitude estimates of how high supersaturation with respect to ice could get around freezing droplets (up to 25 at -15°C ) and then argues that even if that high a supersaturation would pertain throughout the cloud at any time and everywhere this could not explain the factor $10^4$ to $10^5$ of ice number/INP number found in some clouds. His argument relies on a power law for the number of INP as a function of supersaturation, which might be questioned, but has some support. We agree that this is an important area of future research and have modified the respective paragraph:

"However, Baker (1991) argued that even if $T_a$ is as low as -15°C, the total volume with high supersaturation around all freezing drops remains too small to enhance the number concentration of active INP by several orders of magnitude. Therefore, INP activation in transient supersaturation around freezing drops should have a low significance for SIP. This result seems to be conflicting with the conclusion obtained in Rosinski et al. (1975) and Prabhakaran et al. (2020). It should be noted that Baker (1991) assessment of the SIP efficiency was obtained for the static field of supersaturation around droplet and under the assumption, that the number concentration of active INP follows a power law in supersaturation that could be extrapolated to a very high supersaturation." …

We also modified a statement at the end of section 7:

"Future laboratory studies should be focused on the behavior of INP at very high ice supersaturation (>10) for a better the quantification of the effect of $T_s - T_a$ of a free-falling hydrometeor on the INP activation."

13. My initial reaction was that Section 8 does not really fit with the main theme of the review. It is relevant in the sense that spurious ice crystals may have contributed to the apparent conflict between measured INP and measured ice crystal concentrations. But then it begs the question why other field measurements are not reviewed as well. One aspect that could be emphasized to strengthen the connection to the laboratory focus of the paper is that the high speed videos in the Korolev et al. papers were obtained in a wind tunnel setting (at least that is my recollection). Perhaps this is a good place to emphasize that lab "experiments" have contributed not only to understanding of fundamental mechanisms, but also to the evaluation of measurement techniques applied in the field. Those videos captured in a controlled lab environment settled the question of shattered ice crystals in the minds of many in the community. I would also point out that this section misses an important reference to Jackson et al. 2014, JAOTech, who made a full assessment of measurement-induced ice shattering based on intercomparison of multiple instruments.
**Reply**:  The relevant text in section 8 was modified as suggested by the reviewer:
"Direct experimental support for the shattering hypothesis has been provided by a series of wind tunnel experiments with controlled environment and reproducible ice spray conditions (Korolev et al. 2011, 2013b). Ice particle impacting with the probe tips at aircraft speed were video-recorder by a high speed camera."
Jackson et al. JAOT, 2015 reference was added in the text.

14. In Section 9.1 it could be useful to elaborate more on "the most striking outcome of this review", that there is such a wide range of results for each SIP mechanism. In conversation with colleagues I have encountered a sense of exasperation that lab experiments sometimes show bewildering complexity. I even remember a story from an individual involved with experiments in the Hobbs lab in the late '60s that suggested that the CO2 contamination they identified as a cause of drop shattering is one factor that motivated Prof. Hobbs to shift his group's emphasis to field work. The result of the "striking outcome", however, should be that we carry out more, not less experimental work, in order to clarify the various unrecognized factors and ultimately to gain a full understanding of the relevant processes. A thorough check for grammatical and typographical errors should be made. Overall the writing is excellent, but there are multiple places where small errors appear. I summarize the ones I found, although I probably did not catch all while reading:

Title: It sounds more natural to my ear to say "experimental studies of secondary ice production" rather than "on secondary ice production". But I'm not a grammar expert, so I would not go so far as to say it is incorrect.
**Reply**: The authors put their efforts to double check grammar and spelling. The title was modified as suggested by the reviewer.

Line 34: Schaefer
**Reply**: Corrected.

Caption of Figure 4: Delta T = 14.5 C (should not be negative as currently shown).
**Reply**: Corrected.

Line 205 and several other places: A temperature is high or low (not warm or cold, which is only for an object).
**Reply**: Corrected.

Line 208: the Visagie experiments.
**Reply**: Corrected.

Lines 335-336: experiments that had a droplet suspended (no "is" needed).
**Reply**: Corrected.

Caption of Figure 8: I do not find Lauber et al. 2015 in the references. Should it be 2018?
**Reply**: Corrected.

Line 425: number of parameters.
**Reply**: Corrected.

Line 489: in a cloud chamber.
**Reply**: Corrected.

Line 511: studies of Hallett and Mossop.
**Reply**: Corrected.

Line 526: rapid growth of the ice shell.
**Reply**: Corrected

Line 581: Phillips (also this sentence might be clearer is written "The studies of Hobbs and Farber, Vardiman, and Phillips et al. were based on the consideration…").
**Reply**: Corrected.

Caption of Figure 14: AgI and Snomax were used as ice nucleating particles.
**Reply**: Corrected.

Lines 722-723: aerosols introduced into the ambient air.
**Reply**: Corrected.

Line 737: ice sphere in humid air.
**Reply**: Corrected.

Line 738: The word "study" appears twice… should be rephrased.
**Reply**: Corrected.

Line 765: Not sure if "feedbacking" is a word.
**Reply**: Corrected.

Line 783: the existence of shattering.
**Reply**: Corrected.

Line 843: a complete quantitative theoretical description.
**Reply**: Corrected.

Line 848: systematic basis in weather prediction models.
**Reply**: Corrected.

---

## Author Response (AR2)

**Atmospheric Science and Technology Branch**
**4905 Dufferin Street, ARMP**
**Downsview, Ontario**
**M3H 5T4   CANADA**

Atmospheric Chemistry and Physics

27 August 2020

**RE: revised manuscript acp-2020-537**

Dear Daniel,

Thank you very much for careful reading of our revised manuscript and catching for typos and grammar. Below are point-by-point replies on your comments.

Line 536: After "several", I believe a number is missing.
*Reply*: The statement was replaced by the one used in the source work by Macklin (1960): "…varied from few to 140μm…". I believe, that due to the absence of reliable technique for droplet measurements in early 1960, the lower limit of the droplet size distribution was not well established.

Line 537: You want to define "MVD".
*Reply*: "MVD" was expanded to "mean volume diameter"

A suggestion here (not a must): you may consider to give a table with nomenclature for applied parameters (given in text and equations) and important abbreviations. If you think this is not necessary, no need to do it. However, my feeling is that it may benefit the reader.
*Reply*: Appendix A with the table describing symbols was added in the text, as suggested.

line 720 and line 725: Both lines display almost the same sentences. Not sure if this was intended as is. "...collide and nucleate....".
*Reply*: The repeated statement was deleted and two paragraphs were merged into one. Now it reads as follows:
"Dye and Hobbs (1968) observed during laboratory experiments that, when an ice crystal on some occasions became attached to a freezing drop, it would often break into 5 to 10 pieces as the drop froze. Sometimes, the breakup of the crystal would occur when the drop cracked. On other occasions the crystal would break without any apparent changes to the freezing drop. Later Hobbs and Farber (1972) reproduced laboratory experiments of Dye and Hobbs. They observed shattering of a dendritic crystal into several pieces after bringing it in contact with 2mm diameter supercooled drop. These observations are of considerable interest, for it suggests that the breaking up of ice crystals that collide and nucleate supercooled drops, may play an important role in increasing the concentration of ice particles in natural clouds. "

line 977: Did you mean "complete" instead of "compete". Comprehensive? Maybe you can omit it "...none of the above mechanisms have a quantitative theoretical description...". Or: "Although a complete quantitative theoretical description is missing for the above mechanisms, there is a ...".
*Reply*: It should be "complete". Thanks for careful reading. Somehow both co-authors missed this typo after several readings. There are some attempts to provide a theoretical descriptions of different SIP mechanisms. However, they are far from the complete stage.

line 985: "hand" instead of "hang".
*Reply*: Corrected. Thanks for catching it.

The modified manuscript was uploaded at the ACPD site.

Sincerely,

Alexei Korolev,
Research Scientist
e-mail: alexei.korolev@canada.ca